# Biotransformation of polyunsaturated fatty acids to bioactive hepoxilins and trioxilins by microbial enzymes

Jung-Ung An[1], Yong-Seok Song[1], Kyoung-Rok Kim[1], Yoon-Joo Ko[2], Do-Young Yoon[1] & Deok-Kun Oh[1]

Hepoxilins (HXs) and trioxilins (TrXs) are involved in physiological processes such as inflammation, insulin secretion and pain perception in human. They are metabolites of polyunsaturated fatty acids (PUFAs), including arachidonic acid, eicosapentaenoic acid and docosahexaenoic acid, formed by 12-lipoxygenase (LOX) and epoxide hydrolase (EH) expressed by mammalian cells. Here, we identify ten types of HXs and TrXs, produced by the prokaryote *Myxococcus xanthus*, of which six types are new, namely, $HXB_5$, $HXD_3$, $HXE_3$, $TrXB_5$, $TrXD_3$ and $TrXE_3$. We succeed in the biotransformation of PUFAs into eight types of HXs (>35% conversion) and TrXs (>10% conversion) by expressing *M. xanthus* 12-LOX or 11-LOX with or without EH in *Escherichia coli*. We determine 11-hydroxy-eicosatetraenoic acid, $HXB_3$, $HXB_4$, $HXD_3$, $TrXB_3$ and $TrXD_3$ as potential peroxisome proliferator-activated receptor-γ partial agonists. These findings may facilitate physiological studies and drug development based on lipid mediators.

---

[1] Department of Integrative Bioscience and Biotechnology, Konkuk University, 120 Neungdong-ro, Gwangjin-gu, Seoul 05029, Republic of Korea. [2] National Center for Inter-University Research Facilities (NCIRF), Seoul National University, 1 Gwanak-ro, Gwanak-gu, Seoul 08826, Republic of Korea. Correspondence and requests for materials should be addressed to D.-K.O. (email: deokkun@konkuk.ac.kr)

Lipid mediators are signalling transduction molecules essential for homeostasis and intracellular communication in humans. They are C20- and C22-polyunsaturated fatty acids (PUFAs) containing hydroxyl group(s) and/or an epoxide group with or without a C5 ring. Lipid mediators include leukotrienes (LTs), lipoxins (LXs), resolvins (RVs), protectins (PTs), hepoxilins (HXs), trioxilins (TrXs) and prostaglandins (PGs), which are involved in the regulation of the immune and inflammatory responses of humans[1] (see Supplementary Table 1 for abbreviations). Owing to their anti-inflammatory, anti-infective, anti-bacterial, anti-viral, anti-apoptotic, neuroprotective and tissue-healing properties, these lipid mediators have attracted much attention in recent years.

HXs and TrXs are C20- and C22-PUFAs, with the former having a hydroxyl group and an epoxide group at C11 and C12, and the latter having three hydroxyl groups. The classification and chemical names of all HXs and TrXs are presented in Supplementary Table 2. They are found in various types of organs, tissues and cells, including brain[2], aorta[3], insulinoma[4], epidermis[5], platelets[6] and neutrophils[7]. Compounds of HXA series contain a hydroxyl group at C8, whereas compounds of HXB series have a hydroxyl group at C10. In humans, arachidonic acid (ARA) is metabolized to $HXA_3$ and $HXB_3$ by ARA 12-lipoxygenase (LOX), which are converted to $TrXA_3$ and $TrXB_3$ through the hydrolysis of the epoxy group in $HXA_3$ and $HXB_3$, respectively, by HX epoxide hydrolase (EH) (Supplementary Fig. 1). HXs are involved in insulin secretion[8], calcium regulation[9], potassium regulation[10], platelet aggregation[11] and vascular permeability[12]. HXs are also chemotactic factors for human neutrophils like LTs[7] and pathogen-elicited epithelial chemoattractants[13]. HXs and TrXs regulate vasorelaxation in the arteries[14], affect the nuclear receptor peroxisome proliferator-activated receptor alpha (PPARα)[15] and are involved in regulating the life cycle of barnacles, e.g. in egg hatching and larval settlement[16]. Thus, they are important lipid mediators for various organisms.

LOXs, cyclooxygenases (COXs) and the cytochrome P450 families are the starting enzymes for the biosynthesis of lipid mediators using PUFAs as substrates[17]. Among these enzymes, LOXs, a family of non-heme-iron-containing dioxygenases, catalyse the dioxygenation of PUFAs containing one or more $Z,Z$-1,4-pentadiene structures to hydroperoxy fatty acids (HPFAs). They also catalyse the epoxidation of HPFAs to epoxy hydroxy fatty acids (EHFAs) such as LTs and HXs. LOXs are classified as 5-, 8-, 11-, 12- and 15-LOXs according to the number of oxygenated carbon site on ARA. LOXs have been mainly studied in animals and plants, but rarely in other organisms such as corals, fungi and bacteria. EHs, which catalyse the conversion of epoxides into diols with water molecules, are also important enzymes for lipid mediator synthesis, and widely exist in animals, plants, insects and microorganisms[18, 19]. EHs are divided into five types, namely, soluble EH, microsomal EH, EH3, EH4 and HX EH[20]. Although HX EH catalyses the conversion of HXs to TrXs in vivo[21], it has not been used in the biotransformation of PUFAs to lipid mediators.

To date, the lipid mediators LTs, LXs, RVs, PTs, PGs, HXs and TrXs have not been synthesised by recombinant cells expressing microbial enzymes owing to several reasons. First, most enzymes involved in the synthesis of the lipid mediators have originated from mammals. However, mammalian enzymes have significantly lower activities and stabilities towards ARA than those of microbial enzymes. For example, the specific activity of human 12-LOX for ARA (6.78 μmol $min^{-1}$ $mg^{-1}$)[22] was significantly lower (approximately 90-fold) than that of the bacterial 12-LOX from *Myxococcus xanthus* (605 μmol $min^{-1}$ $mg^{-1}$). Second, mammalian enzymes need a eukaryotic host with post-translational modification. However, this host has

disadvantages, such as using of expensive medium, low enzyme expression level and difficulty in scaling-up of production. Third, microbial enzymes involved in the synthesis of the lipid mediators have not been identified yet.

Here, we discover bacterium *M. xanthus* that can produce HXs, TrXs and PGs, and find the biosynthetic genes of HXs, TrXs and PGs from the *M. xanthus* genome. We synthesize diverse HXs and TrXs from PUFAs by expressing genes of *M. xanthus* involved in the biosynthesis of the lipid mediators in *Escherichia coli*. Moreover, the transcriptional activity of PPARγ for HXs and TrXs is determined.

## Results

**LC-MS analysis for ARA-derived metabolites of *M. xanthus*.** *M. xanthus* was cultivated in medium containing ARA for 24 h. After cultivation, the culture supernatant was analysed by high performance liquid chromatography (HPLC) and liquid chromatography-mass spectrometry (LC-MS). *M. xanthus* consumed most of ARA and the peaks of some metabolites were detected (Supplementary Fig. 2). However, ARA was not consumed and there were no new peaks detected in the culture medium without *M. xanthus*. The molecular formulae of metabolites of *M. xanthus* were determined by MS/MS fragmentation analysis (Supplementary Table 3). Seven types of metabolites were suggested by comparison with the references in the LIPID metabolites and pathways strategy (MAPS) Database. They were eicosapentaenoic acid (EPA; C20:5), 11-hydroxy-5Z,8Z,12E,14Z-eicosatetraenoic acid (11-HETE), 12-HETE, 15-HETE, $HXB_3$, $PGG_2$ and $PGH_2$. However, four types of metabolites did not have matched compounds. Among them, metabolite numbers 9 and 10 were possibly new-type of HX and TrX, respectively, since they did not match with the compounds in available information databases, including the LIPID MAPS Database, PubChem, the Human Metabolome Database and KEGG. The other two types of metabolites were not suggested because they had many overlapping MS/MS fragments.

**Identification of the biosynthetic genes and enzymes.** Given that the *M. xanthus* genome has already been sequenced, the eight candidate biosynthetic genes of lipid mediators were selected by comparison with the sequences of human corresponding genes. The genes of *MXAN_1744*, *MXAN_1745*, *MXAN_1644*, *MXAN_5137*, *MXAN_5217*, *MXAN_0683*, *MXAN_2304* and *MXAN_3623* in *M. xanthus* were predicted to be the genes encoding LOX, LOX, EH, EH, COX, two thromboxane A (TXA) synthases and PGD synthase, respectively (Supplementary Table 4). Although the amino acid sequences of these enzymes showed 15–40% identities with human corresponding enzymes[23–29], the major residues affecting the activity were conserved (Supplementary Fig. 3). These candidate genes were cloned and expressed in *E. coli* in soluble forms (Supplementary Fig. 4). No activity was found for putative TXA synthases, putative PGD synthase or putative EH expressed from *MXAN_5137*. The protein from *MXAN_5217* converted ARA to $PGH_2$ (Supplementary Fig. 5), indicating that it is COX. The activity of COX towards ARA was 0.011 μmol $min^{-1}$ $mg^{-1}$. In animals, COX converts ARA to $PGH_2$, which can be converted to diverse PGs by various types of PG synthases (Supplementary Fig. 1). The putative LOX enzymes expressed from *MXAN_1745* and *MXAN_1744*, and the putative EH from *MXAN_1644* were purified from crude cell extracts as single soluble proteins using His-Trap affinity chromatography (Supplementary Fig. 4). The substrate specificity and products of these purified enzymes are summarized in Supplementary Table 5. The enzymes from *MXAN_1745* and *MXAN_1744* converted ARA to 12-

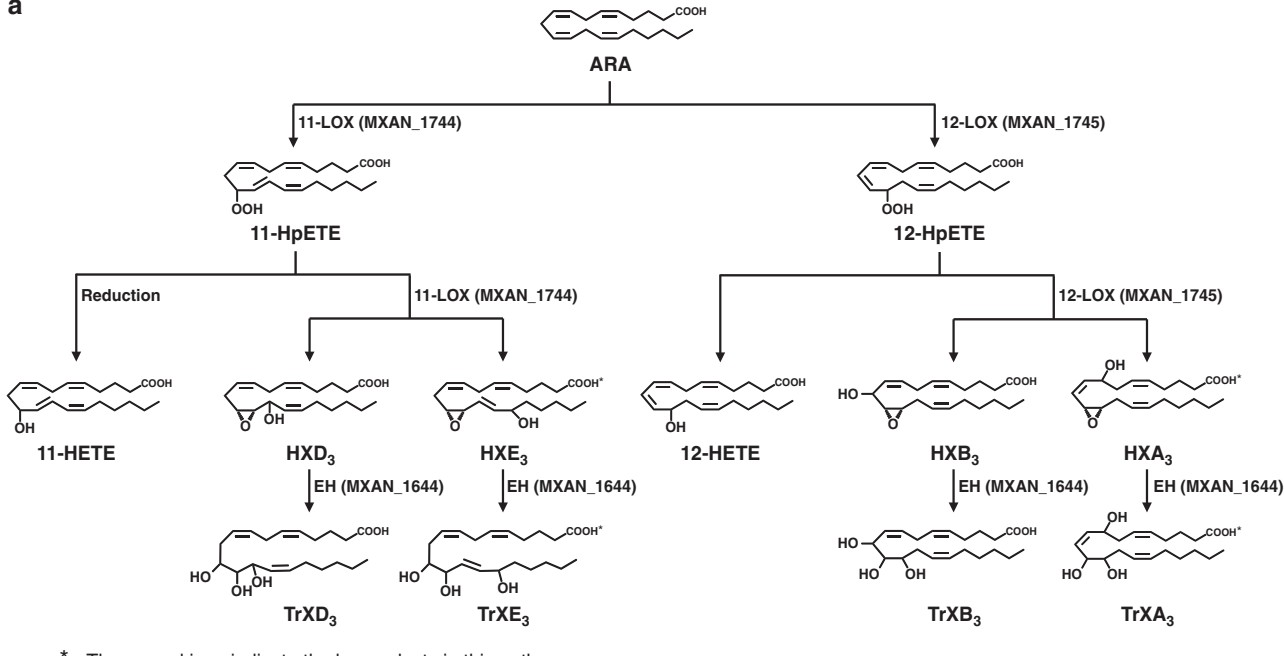

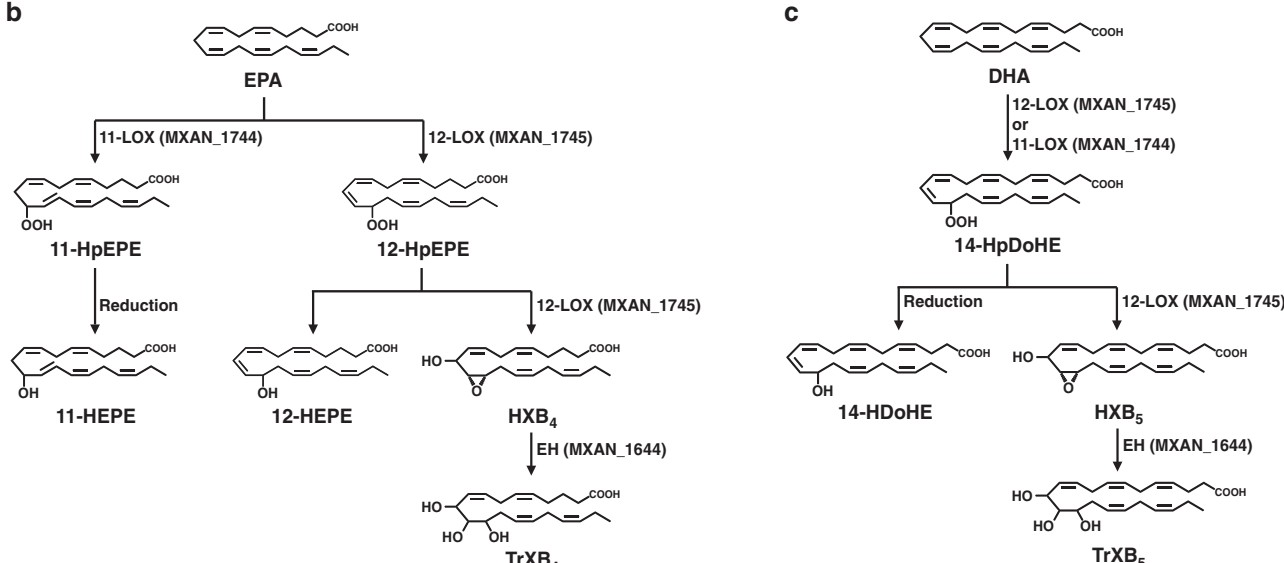

**Fig. 1** Pathways of polyunsaturated fatty acids converting to trioxilins established according to the genes of *Myxococcus xanthus*. PUFA polyunsaturated fatty acid, TrXs trioxilins. **a** Arachidonic acid (ARA) metabolism. **b** Eicosapentaenoic acid (EPA) metabolism. **c** Docosahexaenoic acid (DHA) metabolism

hydroperoxyeicosatetraenoic acid (12-HpETE) and 11-HpETE, respectively, indicating that they are ARA 12-LOX and ARA 11-LOX, respectively. The enzyme expressed from *MXAN_1644* converted $HXB_3$ to $TrXB_3$. Thus, it was identified as EH. The activities of ARA 12-LOX and ARA 11-LOX towards ARA, and EH towards $HXB_3$ were 605, 489 and 1403 $\mu$mol min$^{-1}$ mg$^{-1}$, respectively, which were 55,000, 44,500 and 127,500-fold higher, respectively, than COX activity. COX from *M. xanthus* was not used for the biosynthesis of lipid mediators because of its low activity.

**Establishment of biosynthetic pathways of PUFAs to TrXs.** Although 12-LOX pathways for the conversion of PUFAs to TrXs in humans have already been reported, 11-LOX pathways are not yet known. Recombinant *E. coli* expressing 12-LOX or 11-LOX and EH from *M. xanthus* synthesized HXs and TrXs during

cultivation with ARA for 120 min (Supplementary Fig. 6). However, non-enzymatic products were not found with only ARA and *E. coli* containing ARA in the absence of the plasmid under the same reaction conditions (Supplementary Fig. 7). *E. coli* expressing 12-LOX and EH produced 12-HpETE, 12-HETE, $HXB_3$ and $TrXB_3$, while *E. coli* expressing 11-LOX and EH produced 11-HpETE, 11-HETE, $HXD_3$ and $TrXD_3$. These results suggest that HXs and TrXs can be produced by not only 12-LOX pathways but also new 11-LOX pathways.

To investigate more exactly the biosynthetic pathways for the conversion of ARA to TrXs, the reactions were performed using purified enzymes, including 12-LOX, 11-LOX and EH. We found that 12-LOX and 11-LOX converted ARA to 12-HpETE and 11-HpETE, respectively, and further to $HXB_3$ and $HxD_3$, respectively, which were then converted to $TrXB_3$ and $TrXD_3$, respectively, by EH. 12-HpETE and 11-HpETE were also

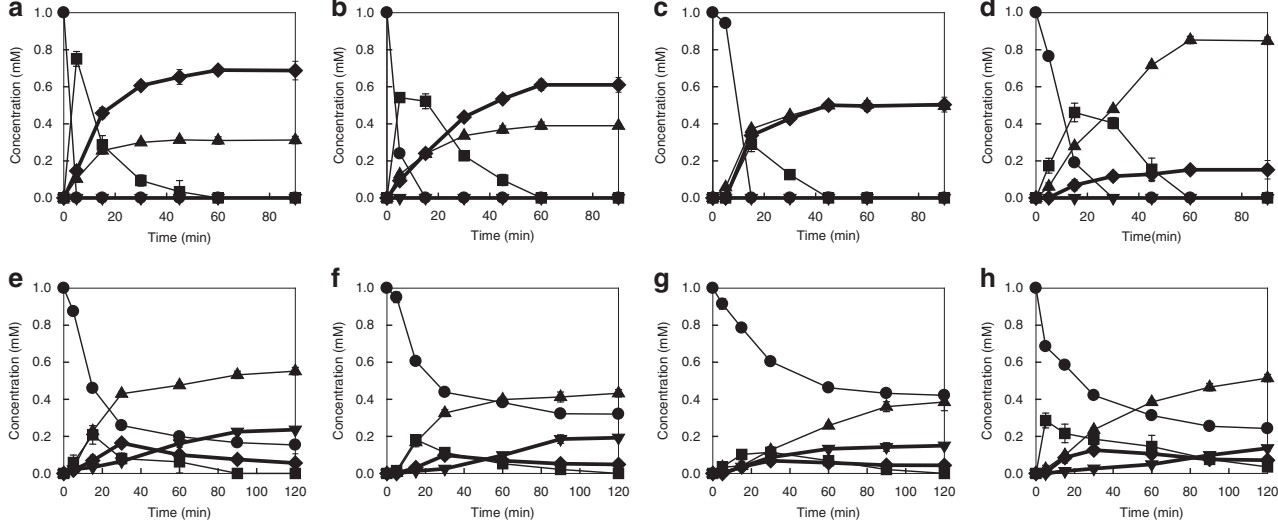

**Fig. 2** Biotransformation of polyunsaturated fatty acids to hepoxilins and trioxilins in *Escherichia coli*. Different combinations of *M. xanthus* genes 12-lipoxygenase (12-LOX) or 11-LOX and epoxide hydrolase (EH) were expressed in *E. coli* strains. HXs hepoxilins, TrXs trioxilins. **a** Biotransformation of ARA to HXB₃ by recombinant *E. coli* expressing 12-LOX. **b** Biotransformation of EPA to HXB₄ by recombinant *E. coli* expressing 12-LOX. **c** Biotransformation of DHA to HXB₅ by recombinant *E. coli* expressing 12-LOX. **d** Biotransformation of ARA to HXD₃ by recombinant *E. coli* expressing 11-LOX. **e** Biotransformation of ARA to TrXB₃ by recombinant *E. coli* expressing 12-LOX and EH. **f** Biotransformation of EPA to TrXB₄ by recombinant *E. coli* expressing 12-LOX and EH. **g** Biotransformation of DHA to TrXB₅ by recombinant *E. coli* expressing 12-LOX and EH. **h** Biotransformation of ARA to TrXD₃ by recombinant *E. coli* expressing 11-LOX and EH. The time-course reactions were performed in 50 mM 4-(2-hydroxyethyl)piperazinyl-1-propanesulphonic acid (EPPS) (pH 8.5) buffer containing 1 mM PUFA, 7.2 g L$^{-1}$ cells for ARA or 14.4 g L$^{-1}$ cells for EPA or DHA at 30 °C for 90–120 min. Data represent the means of three separate experiments, and error bars represent the standard deviations. The symbols indicate PUFA (circles), hydroperoxy fatty acid (HPFA) (squares), hydroxy fatty acid (HFA) (upward-pointing triangles), HX (diamonds) and TrX (downward-pointing triangles)

converted to 12-HETE and 11-HETE by natural reduction, respectively. In particular, 11-LOX produced two types of HXs, HXD₃ and HXE₃. Thus, the 11-LOX and 12-LOX pathways for the conversion of ARA to TrXs were identified (Fig. 1a). We also established the pathways of other eicosanoids, EPA (Fig. 1b) and docosahexaenoic acid (DHA; C22:6), to TrXs using the purified enzymes (Fig. 1c). 12-LOX and 11-LOX converted EPA to 12-hydroperoxypentaenoic acid (HpEPE) and 11-HpEPE, respectively. 11-LOX did not convert 11-HpEPE, whereas 12-LOX converted 12-HpEPE to HXB₄, which was converted to TrXB₄ by EH. 11-LOX and 12-LOX catalysed the same reaction of DHA to 14-hydroperoxydocosahexaenoic acid (HpDoHE). However, only 12-LOX showed epoxidation activity for 14-HpDoHE to HXB₅, because 11-LOX activity was significantly lower than 12-LOX activity. HXB₅ was converted to TrXB₅ by EH.

**Identification of all compounds in the pathways**. Compounds of HXA series (HXA₃[30], HXA₄[31] and HXA₅[32]) and TrXA series (TrXA₃, TrXA₄ and TrXA₅[33]) have already been identified. Although HXB and TrXB series have been reported, their chemical structures have not been identified by nuclear magnetic resonance (NMR). The chemical structures of all compounds involved in the established biosynthetic pathways were suggested by LC-MS/MS analysis (Supplementary Figs. 8–10). The suggested compounds HXB₃, HXB₄, HXB₅, HXD₃, TrXB₃, TrXB₄, TrXB₅, TrXD₃ and TrXE₃ were purified by prep-HPLC (Supplementary Fig. 11). Only *S*-form of 12-HpETE has been used to convert to HX in nature[34]. The 12-HETE and 11-HETE products of *M. xanthus* LOXs were also *S*-forms (Supplementary Fig. 12). The stereoselectivity of HXs and TrXs was suggested and the chemical structures were accurately determined except for TrXE₃ using NMR analysis (Supplementary Tables 6–14 and Supplementary Figs. 13–57). The determination of the structure of TrXE₃ was difficult because the amount produced was very small. Therefore, we just suggested TrXE₃ structure (Supplementary

Fig. 53). HXB₅, HXD₃, HXE₃, TrXB₅, TrXD₃ and TrXE₃ were identified as new compounds, and HXB₃, HXB₄, TrXB₃ and TrXB₄ were first identified by NMR. The detailed explanation for identification of all compounds in the present study was included in Supplementary Notes and Supplementary Methods.

**Biotransformation of PUFAs to HXs and TrXs**. The time-course reactions for the production of HXs and TrXs were performed with 1 mM PUFA or HPFA by recombinant *E. coli*. *E. coli* expressing 12-LOX converted 1 mM of ARA, EPA and DHA to 0.68 mM HXB₃, 0.61 mM HXB₄ and 0.50 mM HXB₅, respectively, in 90 min, with molar conversions of 68%, 61% and 50%, respectively (Fig. 2); converted 1 mM of 12-HpETE, 12-HpEPE and 14-HpDoHE as intermediates to 0.76 mM HXB₃, 0.51 mM HXB₄ and 0.53 mM HXB₅, respectively, in 60 min (Supplementary Fig. 58); and produced 2.15 mM HXB₃ from 6 mM ARA after 60 min, with a conversion of 36% (Supplementary Fig. 59). *E. coli* expressing 11-LOX converted 1 mM ARA to 0.15 mM HXD₃ in 90 min and converted 1 mM 11-HpETE to 0.27 mM HXD₃ in 60 min (Fig. 2d and Supplementary Fig. 58d). *E. coli* co-expressing 12-LOX and EH converted 1 mM of ARA, EPA and DHA to 0.23 mM TrXB₃, 0.19 mM TrXB₄ and 0.14 mM TrXB₅, respectively, in 120 min (Fig. 2e–g), and *E. coli* co-expressing 11-LOX and EH converted 1 mM ARA to 0.13 mM TrXD₃ in 120 min (Fig. 2h).

**Determination on the transcriptional activity of PPARγ**. PPARγ, a type II nuclear receptor, regulates fatty acid storage and glucose metabolism. Its agonists have been used in the treatment of hyperlipidaemia and hyperglycaemia. The effects of HETEs, HXs and TrXs on the transcriptional activity of PPARγ were investigated to find PPARγ agonists. The effects of HXB₃, HXB₅ and HXD₃ on the transcriptional activity of PPARγ were similar to those of their corresponding TrXs. HXB₃ and HXD₃ (TrXB₃ and TrXD₃) increased the transcriptional activity of PPARγ with increasing concentrations, although the increasing degrees of the

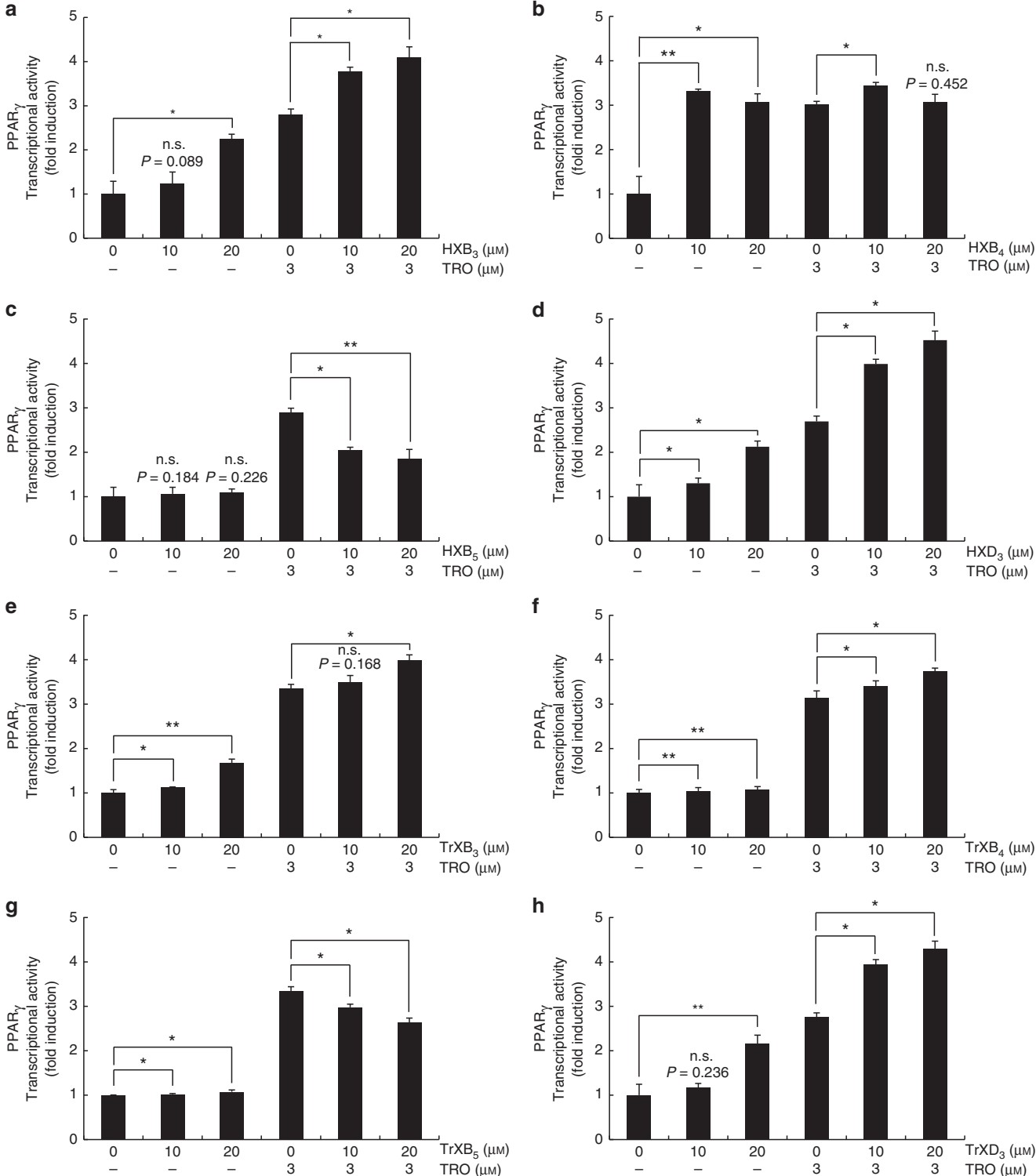

**Fig. 3** Transcriptional activity of peroxisome proliferator-activated receptor gamma for hepoxilins and trioxilins. HEK-293 cells were cultured in a 24-well plate ($1.0 \times 10^5$ cells per well). After 24 h, cells were transfected with plasmids expressing peroxisome proliferator-activated receptor gamma (PPARγ), PPAR response element (PPRE)×3-thymidine kinase-luciferase reporter constructs, and the *Renilla* luciferase control vector pRL. After another 24 h, cells were treated with HXs, TrXs and/or 3 μM troglitazone (TRO) for 24 h. Cells were harvested, and the transcriptional activity of PPARγ was determined by a luciferase assay. HXs hepoxilins, TrXs trioxilins. **a** HXB₃. **b** HXB₄. **c** HXB₅. **d** HXD₃. **e** TrXB₃. **f** TrXB₄. **g** TrXB₅. **h** TrXD₃. Data represent the means of three separate experiments, and error bars represent the standard deviations. *p*-value are based on *t*-test. *$p < 0.05$, **$p < 0.01$. n.s. indicates not significant

transcriptional activity were less than that of troglitazone (TRO), an antidiabetic and anti-inflammatory drug (Fig. 3). These compounds additively increased the transcriptional activity of PPARγ when TRO was supplemented. The increasing degree of the transcriptional activity for $HXB_4$ was similar to that by TRO (Fig. 3b). $HXB_5$, $TrXB_4$ and $TrXB_5$ did not affect the activity (Fig. 3c, f, g). However, $HXB_5$ decreased the transcriptional activity of PPARγ when TRO was supplemented. 11-HETE and 12-HETE as intermediate products showed effects similar to those of $HXB_3$ and $HXB_5$, respectively, on the transcriptional activity of PPARγ (Supplementary Fig. 60).

## Discussion

Lipid mediators regulate the immune and inflammatory responses of humans. LOXs are key enzymes involved in the formation of lipid mediators in animals and humans[35]. Recently, LOXs have been discovered in diverse organisms such as coral[36], fungi[9, 37] and bacteria[38]. Among them, bacterial LOXs have only been found in the cyanobacteria *Nostoc* sp.[39] and *Acaryochloris marina*[40], and the proteobacteria *Pseudomonas aeruginosa*[41] and *Burkholderia thailandensis*[38] (Supplementary Fig. 61 and Supplementary Table 15). *M. xanthus* is also a proteobacterium. Interestingly, LOXs in other proteobacteria have regiospecificity on C15 of ARA, whereas LOXs in *M. xanthus* has regiospecificity on C12 and C11 of ARA. Bacteria containing LOXs may produce lipid mediators. However, the formation of HXs and TrXs by bacterial LOXs has not been reported thus far.

In humans, HX is formed from ARA via 12-HpETE by dioxygenation and epoxidation reactions of 12-LOX[42]. $HXA_3$[9] and $HXB_3$[30], $HXA_4$ and $HXB_4$[31], and $HXA_5$[32] are formed from ARA, EPA and DHA, respectively. Then, these HXs are converted to $TrXA_3$, $TrXB_3$, $TrXA_4$, $TrXB_4$ and $TrXA_5$, in vivo by EH, respectively (Supplementary Fig. 62). 15-LOX also converts ARA to HX analogues[43]. $TrXC_3$ is specifically produced from $HXA_3$[33]. In this study, 12-LOX from *M. xanthus* converted ARA, EPA and DHA to $HXB_3$, $HXB_4$ and $HXB_5$, respectively, which were further converted to $TrXB_3$, $TrXB_4$ and $TrXB_5$, respectively, by EH from *M. xanthus* (Fig. 1). Among them, $HXB_5$ and $TXB_5$ were the new products. 11-LOX, catalysing dioxidation on C11 in ARA, has not been reported in animals but exists in the cyanobacterium *A. marina* and some algae[40, 44]. 11-LOX discovered in the proteobacterium *M. xanthus* converted ARA to new HXs with hydroxyl groups located at C13 and C15 (Fig. 1a). We named these HXs as $HXD_3$ and $HXE_3$, respectively, which were converted to the new compounds $TrXD_3$ and $TrXE_3$, respectively, by EH. The experiments for the presence or absence of these compounds in mammalian tissues using $^{14}C$-labeled ARA were performed by other research groups[5, 32]. As a result, $HXD_3$ and $TrXD_3$ were not found in human and rat tissues. Thus, not only 11-LOX but also 11-LOX-derived products such as $HXD_3$ and $TrXD_3$ do not exist in mammalian tissues.

We found the lipid mediators HXs, TrXs and PGs in the culture broth containing ARA and *M. xanthus* (Supplementary Table 3), and identified 12-LOX, 11-LOX, EH and COX as the related enzymes in *M. xanthus* (Supplementary Table 4). HXs and TrXs were produced from PUFAs by *E. coli* expressing these bacterial enzymes. The production is our unprecedented discovery. The availability of HXs and TrXs has been strictly limited because there is only one reagent-grade $HXA_3$ commercially available. However, recombinant cells in the present study converted 1 mM PUFAs to four types of HXs with more than 50% conversion (Fig. 2a–d) and four types of TrXs with more than 10% conversion (Fig. 2e–h). These conversion rates may be sufficient for the economical production of diverse HXs and TrXs, which opens the door for physiological studies and drug development.

PPARγ is an important nuclear receptor functioning in lipid accumulation[45], glucose metabolism[46], inflammatory response[47], neutrophil transmigration[48], vascular permeability[49] and hyperalgesia[50]. Thiazolidinediones (TMZs) such as rosiglitazone[51] and TRO[52] are representative full agonists of PPARγ. Full agonists have side effects such as weight gain and worsening of congestive heart failure[53], whereas partial agonists retain beneficial antidiabetic properties with reduced side effects[34]. Partial agonists are defined as weak activators of PPARγ that show the same activation pattern with lower transactivation potential compared to full agonists[54]. The ligand binding domain (LBD) of PPARγ consists of a bundle of 13 α-helices and 4 β-strands, and agonists are bind to helices H3, H5, H10 and H12, including the major residues Ser289 (H3), His323 (H5), His449 (H10) and Tyr473 (H12)[55]. Full agonists of PPARγ such as TMZs are known to bind to H12, whereas partial agonists stabilize the H2′/H3 and H5 areas, resulting in distinct transcriptional effects between full and partial agonists[25]. Thus, the additive effects of full and partial PPARγ agonists exist[56]. In this study, the transcriptional activity of PPARγ in response to the ten types of lipid mediators produced by *M. xanthus* enzymes was investigated (Fig. 3 and Supplementary Fig. 60) because fatty acids have played a role as modulators of PPARγ[57]. $HXB_3$, $HXB_4$, $HXD_3$, $TrXB_3$, $TrXD_3$ and 11-HETE increased the transcriptional activity of PPARγ. These compounds were docked to LBD at human PPARγ using molecular models (Supplementary Fig. 63). Rosiglitazone, known as a full agonist, was interacted with Tyr473 (H12) (Supplementary Fig. 64a). $HXB_3$, $HXB_4$, $HXD_3$, $TrXB_3$, $TrXD_3$ and 11-HETE were interacted with Ser289 (H3) and His323 (H5) (Supplementary Fig. 64b–g), suggesting that these compounds are partial agonists. However, this study does not demonstrate direct agonist activities of these products.

In conclusion, we discovered eukaryotic-like lipid mediator-biosynthetic enzymes, including 12-LOX, 11-LOX, COX and EH, from *M. xanthus*, a newly discovered bacterium that could produce HXs, TrXs and PGs. Owing to the high activities of microbial enzymes, we succeeded in the biotransformation of PUFAs to eight types of HXs and TrXs using recombinant cells expressing 12-LOX or 11-LOX with or without EH. The same strategy can be applied to the biotransformation processes of PUFAs to other lipid mediators such as LTs, LXs, RVs and PTs. We identified ten types of lipid mediators including six new types based on NMR analysis. We found that six types of lipid mediators were potential partial agonists of PPARγ. The identification of partial agonists of PPARγ has been required for development of the antidiabetic and anti-inflammatory drugs with reduced side effects. Thus, outcomes of this study may hold potential to stimulate physiological studies and drug development on lipid mediators.

## Methods

**Materials**. The PUFA standards ARA, EPA and DHA, and the HFA standards 11-HETE, 12-HETE, 12-HEPE and 14-HDoHE were purchased from Sigma (St. Louis, MO, USA) and Cayman Chemical (Ann Arbor, MI, USA), respectively. To prepare the lipid mediator standards $HXB_3$, $HXB_4$, $HXB_5$, $HXD_3$, $TrXB_3$, $TrXB_4$, $TrXB_5$ and $TrXD_3$, the reactions were performed at 30 °C in 50 mM 4-(2-hydroxyethyl)piperazinyl-1-propanesulphonic acid (EPPS) buffer (pH 8.5) containing 100 mg L$^{-1}$ of ARA, EPA or DHA as a substrate, and 14.4 g L$^{-1}$ recombinant cells with shaking at 200 r.p.m. for 2 h. The reaction solution was extracted with an equal volume of ethyl acetate, and the solvent was removed using a rotary evaporator. The solvent-free solution was applied to a Prep-HPLC (Agilent 1260, Santa Clara, CA, USA) equipped with a Nucleosil C18 column (10×250 mm, 5-μm particle size; Phenomenex, Torrance, CA, USA) and a fraction collector. The column was eluted at 30 °C with a flow rate of 6 mL min$^{-1}$, and the product fractions were collected by monitoring at 202 nm of absorbance. The collected samples showed >99% purity (Supplementary Fig. 11), and were used as the lipid mediator standards after identification by LC-MS/MS and NMR.

**Plasmids and microorganisms culture conditions.** *M. xanthus* KCCM 44251 (Korea Culture Center of Microorganisms, Seoul, Republic of Korea), *E. coli* BL21, and pET-28a and pACYC duet plasmids were used as the source of genomic DNA, host cells and expression vectors, respectively. *M. xanthus* was cultivated in a 500-mL flask containing 100 mL of Casitone medium supplemented with 1 mM ARA at 30 °C with shaking at 200 r.p.m. for 24 h. Recombinant *E. coli* was cultivated in a 2-L flask containing 450 mL of Luria−Bertani (LB) medium supplemented with 20 µg mL$^{-1}$ kanamycin for pET-28a or chloramphenicol for pACYC duet vector at 37 °C with shaking at 200 r.p.m. When the optical density of the bacterial culture at 600 nm reached 0.6, 0.1 mM isopropyl-β-D-thioglactopyranoside was added, and the incubation continued with shaking at 150 r.p.m. at 16 °C for 18 h to induce enzyme expression.

**Gene cloning.** The genes encoding candidate enzymes were amplified by PCR using *M. xanthus* genomic DNA as a template (Supplementary Table 4). The primers used for gene cloning were designed based on the DNA sequence of candidate enzymes from *M. xanthus* (Supplementary Table 16). DNA fragments obtained by PCR amplification with *Taq* polymerase (Solgent, Daejon, Korea) were ligated into the pET-28a or pACYC duet vector. The resulting plasmid was transformed into *E. coli* BL21 and then plated on LB agar containing 20 µg mL$^{-1}$ kanamycin (pET 28a vector) or chloramphenicol (pACYC duet vector). An antibiotic-resistant colony was selected, and the plasmid DNA was sequenced using a DNA analyser (ABI Prism 3730xl; Perkin-Elmer, Waltham, MA, USA).

**Enzyme purification.** Harvested cells were suspended in 50 mM phosphate buffer (pH 8.0) containing 10 mM imidazole, 300 mM NaCl and 0.1 mM phenylmethylsulphonyl fluoride as a protease inhibitor, and disrupted by sonication on ice bath. Cell debris was removed by centrifugation at 13,000×*g* for 10 min at 4 °C, and the supernatant was applied to an immobilized metal ion affinity chromatography cartridge (Bio-Rad, Hercules, CA, USA) equilibrated with 50 mM phosphate buffer (pH 8.0) containing 300 mM NaCl. The bound protein was eluted by the same buffer with a linear gradient of 10−250 mM imidazole at a flow rate of 1 mL min$^{-1}$. The active fractions were collected and loaded onto a Bio-Gel P-6 desalting cartridge (Bio-Rad) equilibrated with 50 mM EPPS buffer (pH 8.5). The loaded protein was eluted using the same buffer at a flow rate of 1 mL min$^{-1}$, and the eluted protein was used as the purified enzyme.

**Enzyme and cell reactions.** To measure the specific activities of the enzymes, the reactions were performed at 30 °C in 50 mM EPPS (pH 8.5) containing 1 mM substrate and 0.12−2.0 g L$^{-1}$ of 12-LOX, 11-LOX, EH or COX for 5 min. The reactions for the conversion of PUFAs into lipid mediators were performed at 30 °C in 50 mM EPPS (pH 8.5) containing 1−6 mM substrate and 3.6−14.4 g L$^{-1}$ cells for 120 min. The optical density at 600 nm of the cell suspension was measured and converted to dry cell weight.

**Transcriptional activity assay.** Human embryonic kidney (HEK) 293 cells have been widely used for the screening of PPARγ agonists[58], suggesting that the endogenous factor of HEK293 cells do not affect the transcriptional activity of PPARγ. The lipid mediators tested were not metabolized by HEK 293 cells (Supplementary Fig. 65). To investigate the effects of lipid mediators on the transcriptional activity of PPARγ, HEK 293 cells were cultivated in 24-well plates (1×10⁵ cells per well) containing Dulbecco's modified Eagle's medium with 10% foetal bovine serum for 24 h. Cultured cells were transfected with plasmids expressing PPARγ and PPAR response element×3-thymidine kinase-luciferase reporter constructs (1 µg per well) using the transfection reagent Lipofectamine 2000 (Invitrogen, Carlsbad, CA, USA). After 24 h, cells were treated with 5−20 µM lipid mediator and/or 3 µM TRO. The high micromolar concentration of HETE, HX or TrXs up to 20 µM was used to determine the transcriptional activity of PPARγ because there was no cytotoxicity at this concentration. The concentration up to 20−40 µM has been used for PPARγ partial agonists due to their weak activity and no cytotoxicity[59–61]. The concentration of the full agonist TRO was the same as those used in other reports[47, 62]. After another 24 h, harvested cells were assayed with a dual-luciferase reporter gene assay kit (Promega, Madison, WI, USA). The activities are presented as the expression ratio of firefly luciferase to *Renilla* luciferase.

**Phylogenetic analysis.** 12-LOX and 11-LOX from *M. xanthus* were used as query sequences for a blast search against the genomic sequences of other organisms. All hits with an expected value of <*e*$^{-10}$ were compiled from the database and aligned using the MUSCLE algorithm in MEGA 6. Sequences with poor alignment and annotated as unrelated proteins were removed. Phylogenetic trees were built using the neighbour-joining method in MEGA 6 with 1000 bootstraps.

**HPLC quantitative analysis.** All compounds were quantitatively analysed using an HPLC system (Agilent 1260) with a reversed-phase Nucleosil C18 column (3.2×150 mm, 5-µm particle size; Phenomenex). Absorbance at 202 and 234 nm has been used in HPLC analysis for monitoring non-conjugated HFAs, including HXs, TrXs and PGs, and absorbance at 234 nm has been used for monitoring

conjugated HFAs[63]. HXA$_3$ has been also monitored at 254 nm[64]. HXs and TrXs were detected at a wavelength of 202 nm but not at 234 and 254 nm (Supplementary Fig. 66). Thus, all products were monitored at 202 nm. The column was eluted at 30 °C with 100% solvent A (acetonitrile/water/acetic acid, 50:50:0.1, v/v/v) at a flow rate of 0.25 mL min$^{-1}$ for 0−5 min, solvent A to solvent B (acetonitrile/acetic acid, 100:0.1, v/v) for 5−21 min at 0.25 mL min$^{-1}$, 100% solvent B at 0.4 mL min$^{-1}$ for 21−27 min, solvent B to solvent A at 0.4 mL min$^{-1}$ for 27−32 min, and 100% solvent A at 0.25 mL min$^{-1}$ for 32−35 min. The reaction products were identified to have the same retention times as those of their corresponding standards. The concentrations of PUFAs, HPFAs, HFAs, HXs and TrXs were calculated by calibrating the peak areas to the concentrations of standards. For an example, the determination method for the concentration of HXB$_3$ using a calibration curve was provided in Supplementary Fig. 67.

**In silico docking studies.** Metabolites were docked in the LBD of crystal structure of human PPARγ (PDB 2PGR.pdb) using the CDOCKER module of Discovery Studio 4.1 (Accelrys, San Diego, CA, USA). Substrate poses were refined by full-potential final minimization, and candidate poses were created using random rigid-body rotations followed by simulated annealing. The structure of protein−ligand complexes was subjected to energy minimization using the CHARMM force field in DS 4.5. The substrate orientation with the lowest interaction energy was selected for the subsequent rounds of docking. Candidate poses were created based on random rigid-body rotations followed by simulated annealing. The energy-docked conformation of the substrate was retrieved for post-docking analysis using the CDOCKER module.

**Statistical analyses.** The means and standard errors for all experiments were quantitatively calculated with *t*-test to evaluate significant differences between control and experimental groups. A *p*-value of <0.05, calculated using *t*-test, was considered statistically significant.

**Data availability.** Plasmids used in this article were deposited in Addgene. They were assigned to pET28a-mxLOX1 (ID 104975), pET28a-mxLOX2 (ID 104976), pET28a-mxEH (ID 104977), pACYCduet-mxLOX1-EH (ID 104978) and pACYCduet-mxLOX2-EH (ID 104979). All data that support the findings of this study are included in this article and in Supplementary Information. They are available from the corresponding author upon request.Ed.: Please confirm the plasmid IDs you deposited in Addgene are publicly available.Now, we are prepared our plasmids for deposition and will send to Addgene within this week. Therefore, the plasmid IDs will be available soon.

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

## Acknowledgements

This study was funded by the Mid-Career Researcher Program, through the National Research Foundation grant funded by the Ministry of Science, ICT and Future Planning, Republic of Korea (No. 2016R1A2B3006881).

## Author contributions

D.-K.O. supervised this study. J.-U.A. performed most experiments, including cloning, protein expression, enzyme purification, and enzyme and cell reactions. Y.-S.S. carried out the transcriptional activity assay, and Y.-J.K. analysed the NMR data. J.-U.A., Y.-S.S., K.-R.K., D.-Y.Y., and D.-K.O. designed the experiments and contributed to data analysis. J.-U.A. gathered and organised the results. J.-U.A., Y.-S.S., K.-R.K., Y.-J.K., D.-Y.Y., and D.-K.O. wrote the manuscript.

## Additional information

**Competing interests:** The authors declare no competing financial interests.

