## [Peer Review File · Nature Communications]

Reviewers' comments:

Reviewer #1 (Remarks to the Author):

The manuscript entitled “Biotransformation of polyunsaturated fatty acids to hepoxilins and trioxilins, human lipid mediators, by microbial enzymes” presents results demonstrating that *M. xanthus* expresses enzymes that can carry out lipoxygenase-like reactions to generate hepoxilins, trioxilins and monohydroxy eicosanoids. While the identification of putative lipoxygenase enzymes in *M. xanthus* is important, several experimental problems preclude definitive identification of enzymatic formation of hepoxilins and trioxilins. Specific comments and suggestions that may help to improve the results and interpretation are given below:

1. The Results section lists 10 metabolites that were generated from *M. xanthus*, as identified by LC-MS/MS and matched using the Lipid MAPS database. The present database contains MS/MS spectra for only 2 hepoxilins: HXA3 and HXB3, and one trioxilin (TrXA3). The MS/MS spectra for HXB3 in the Lipid MAPS structure database looks very different from that given in Supplemental Figure 4 in terms of the diagnostic fragmentation ions observed. The majority of the major fragment ions (i.e., 134, 137, 151, 153, 163, 183, 263, and 289) are missing. This should be clarified. Moreover, Supplementary Table 3 lists HXD3 and TrXB3 as being matched in the Lipid MAPS database, although MS/MS spectra for these are not available in the database and the text states that HXD3 was not matched and is therefore a novel structure. These points should be clarified.
2. Regarding point 1 above, several MS/MS spectra are given for the hepoxilins and trioxilins without reference to the identity of the specific fragment ions. If these have been empirically determined, the diagnostic fragment assignments should be provided. If they have not been determined, this should be stated clearly and some attempt to assign these fragments should be made.
3. The products were isolated by HPLC by monitoring absorbance at a wavelength of 202 nm. This type of general analysis is not defined for chromophores specific to the proposed structures. Therefore, the chromatogram is not that helpful and should be replaced by something more informative. For example, formation of a conjugated diene in monohydroxy eicosanoids could be monitored at a wavelength of ~235nm, etc...

4. The authors incubated *M. xanthus* in media containing arachidonic acid for 24 hours and measured the products by LC-MS/MS. They should provide levels of these mediators identified from media alone, as non-enzymatic products could be readily formed under these conditions. The chromatography methods employed would likely not discriminate between stereoisomers and thus it is not clear if the products are racemates. The lack of stereochemically pure standards impedes unequivocal discrimination of enzymatic products from non-enzymatic products. That 11-HETE is one of the major products and is known to be formed non-enzymatically, reinforces this point. Moreover, the authors have named an uncharacterized enzyme based on the presence of 11-HETE (e.g. 11-LOX), which as noted above, can be formed through non-enzymatic oxidation and also via cyclooxygenase (COX) enzymes in low amounts. Notably, the authors also identified COX homologs in *M. xanthus*. Thus, it is not clear that naming this enzyme “11-LOX” is justified and appropriate at this juncture.

5. Following from above, to my knowledge, there is no mammalian homolog of “11-LOX” and thus it is unclear if the products formed (i.e., the novel HXD3 and TrXD3) from this bacterial culture would be present or of physiological relevance in human tissue.

To this end, for novel structures encountered from the bacteria, the authors should determine whether these products are present in mammalian tissues. This analysis would be aided by the biogenic standards they have isolated and could be significantly enhanced by using deuterium-labeled precursors (ie., arachidonic acid) to generate labeled products for use as internal standards.

6. The authors used preparative HPLC to purify biogenic compounds and subsequently performed 2D-NMR to determine their structures. However, very little information is given in the text regarding the structures. Moreover, there is no mention of the stereochemistry given in the text and whether stereochemically pure products were obtained.

7. Regarding the conversion of substrate to product in *E. coli* expressing the *M. xanthus*-derived enzymes, a comparative quantification of the hepxilins and trioxilins with and without the presence of epoxide hydrolase (EH) should be shown in a combined graph. At present, hepxilin levels are only provided in *E. coli* incubations without the presence of EH and trioxilin production is only quantified in bacteria with EH. Was conversion 100% complete? To understand these product-precursor relationships, all compounds should be measured and presented for each incubation.

8. Following from above, the authors never provide levels of any of the lipid mediators in *E. coli* in absence of expression of the enzymes derived from *M. xanthus*, making it difficult to connect biosynthesis solely to expression of the enzymes (again, some of these products could be formed non-enzymatically during the incubations with *E. coli*).

9. With the *E. coli* or isolated enzyme incubations (e.g., Figs 2 and 3), it is not clear from the methods how the products were quantified at each time point. Was an aliquot collected and subjected to LC-MS/MS at each time point? How was the accurate concentration of each product determined? Were molar extinction coefficients empirically determined for the new products?

10. The authors present results suggestive of agonist activity of the hepoxilins and trioxilins for PPAR γ . However, high micromolar concentrations were used for these assays and it is not clear how these concentrations were justified (see Point 9 above). Moreover, there is no statistical analysis presented to determine whether the increase in activity is significant or not. Overall, the PPAR γ activity assay seems to be completely out of place. The physiological relevance of this is not at all clear. Given the documented pro-inflammatory roles of hepoxilins, such as promoting neutrophil transmigration, vascular permeability, and hyperalgesia, it is not clear why activation of PPAR γ has been selected as the sole readout of the potential biological activity of these products. The suggestion that these new products “can be developed as new generation drugs instead of conventional agonists, TMZ’s” ..”because of known side effects”, is a highly speculative statement and should be removed.

11. Results are presented of incubations with isolated purified enzymes (i.e., 12-LOX and so-called 11-LOX of bacterial origin). However, no information is given regarding the purity of the enzyme preparations.

12. For the comparative sequence analysis, the authors should provide information about the alignment with the human enzymes and % sequence similarity.

Reviewer #2 (Remarks to the Author):

In this manuscript, the authors report on the identification of 12-LOX, 11-LOX and EH genes in the bacterium *Myxococcus xanthus*. It is the first bacterium identified as producing lipid mediators so far known in mammals. They identified these polyunsaturated fatty acid metabolites of *M. xanthus* by LC-MS analysis. This is an important finding as it is the first report on the formation of hepoxilins (HXs), trioxilins (TrXs) and prostaglandins (PGs) by bacteria.

Eight candidate biosynthetic genes of lipid mediators from *M. xanthus* were selected by sequence comparison with those of human lipid mediators. Three of these genes (MXAN_1745, MXAN_1744 and MXAN_1644) were used to obtain recombinant *E. coli* expressing 12-LOX or 11-LOX and EH from *M. xanthus*. This led to the discovery of new 11-LOX pathways for the conversion of arachidonic acid (ARA) to TrXs and the conversion of the

polyunsaturated fatty acid HDA to TrXB5. In addition, new compounds of these enzymatic activities were identified. Several of human lipid mediators produced by *M. xanthus* enzymes were investigated for the activation of the Peroxisome Proliferator Activated Receptor (PPAR) γ . HXB3, HXD3, TrXB3, TrXD3, and 12-HETE demonstrated PPAR γ activity upregulation. However, they were less effective than that thiazolidinedione Troglitazone. In addition to the new knowledge generated by this study, the possibility of a microbial production of human lipid mediators in relatively important amounts will facilitate the investigation of their physiological roles and encourage the development of new PPAR γ drugs potentially devoid of the side effects caused present thiazolidinediones.

A large amount of data is presented and the findings reported advance the field, especially on the compounds produced by the *M. xanthus* enzymes.

The less convincing part of the study is the chapter on "Determination of the effect of human lipid mediators on PPAR γ transcriptional activity compared with an antidiabetic and anti-inflammatory drug", which raises several questions:

- 1) There is no statistical analysis of the results presented in Figure 4.
- 2) What is the evidence that the compounds tested are not metabolized by the human embryonic kidney (HEK) cells? Metabolites of the compounds may be the active substances.
- 3) Are the compounds that increase the transcriptional activity of PPAR γ bona fide PPAR γ ligands? An *in silico* docking analysis could have been done which show (or not) that the compounds can fit into ligand binding pocket of PPAR γ . In addition, although technically more demanding, would be *in vitro* experiments testing the ability of the active compounds to recruit a co-activator to the ligand binding domain.
- 4) Figure 4. Troglitazone increases the transcriptional activity of PPAR γ by 2.5 to 3.5 fold. In Figure 4b, the activation is very weak indicating that this experiment has not worked well, but lead to the conclusion that HXB4 is not an activator. If the control has not worked well, this conclusion may be wrong.
- 5) For active compounds (HXB3, HXD3, TrXB3, TrXD3) and additive effect if observed when they are tested together with Troglitazone. How is this being explained? Have experiments been done with lower or higher concentrations than 3 μ M of Troglitazone?
- 6) The highest concentration tested for compounds is 20 μ M. For active compounds there is no evidence that the maximum effect is reached at this concentration.

Reviewer #3 (Remarks to the Author):

The manuscript NCOMMS-17-13721, deals with the production of hepoxilins and trioxilins by

microbial enzymes. Although this is to some extent novel, several major methodological issues are apparent:

1. The authors incubated ARA for 24 hours with the bacteria. How did the authors control for autoxidation phenomena? Why is no data of control incubations presented? Did the authors investigate the employed ARA batch for the presence of autoxidation products? As very high amounts of PUFA were used it is mandatory to test the starting materials for impurities, autoxidation signals, has this been tested?
2. The presented MS/MS spectra seem very clean to me, why is there basically no noise? Was this such high concentrations used to obtain these data? Was this data obtained from pure standards or incubations?
3. The NMR data which apparently was used for the accurate structural elucidation seems inappropriate. The authors specify double bond geometry as well as stereochemistry. I am wondering how the authors could define absolute stereochemistries? Using NMR one would need to label with Mosher's acid chloride or use shift reagents, this is not apparent from the manuscript. Alternatively chiral chromatography and comparison with standards would also suffice or the use of circular dichroism, again, maybe I missed it but this is not obvious from the text. Next, the authors give double bond geometries, however the NMR tables in the supplement do not contain the coupling constants for the double bond protons, which would be important to judge the E/Z geometry. Alternatively I would have seen DQF-COSY or J-res and NOESY as possible methods to specify the coupling constants and the geometry. The authors give a ROESY spectrum image which actually looks like a NOESY but without any information about peak integrals (or intensities) that could assist the analysis of stereochemistry. Furthermore, this spectrum image clearly contains TOCSY peaks too, and thus, it is not clear how the authors determined the stereochemistry using this data.
4. The authors also state to have identified lipid mediators by comparison with Lipidmaps. This is inappropriate. Substances like PGE₂ and PGH₂ give similar tandem mass spectra and can only be identified taking retention times into account. Again it is not obvious to me that this has been done.

Over all I am sorry but I have to pledge for rejection of the manuscript in its present form.

Responses to Reviewers:

Response to Reviewer #1

1. (1) The Results section lists 10 metabolites that were generated from *M. xanthus*, as identified by LC-MS/MS and matched using the Lipid MAPS database. The present database contains MS/MS spectra for only 2 hepoxilins: HXA₃ and HXB₃, and one trioxilin (TrXA₃). (2) The MS/MS spectra for HXB₃ in the Lipid MAPS structure database looks very different from that given in Supplementary Figure 4 in terms of the

diagnostic fragmentation ions observed. The majority of the major fragment ions (i.e., 134, 137, 151, 153, 163, 183, 263, and 289) are missing. This should be clarified. (3) Moreover, Supplementary Table 3 lists HXD₃ and TrXB₃ as being matched in the Lipid MAPS database, although MS/MS spectra for these are not available in the database and the text states that HXD₃ was not matched and is therefore a novel structure. These points should be clarified.

Answer) (1) The expression of “10 metabolites matched using the Lipid MAPS database” was incorrect. It was our mistake. 8 metabolites matched using the Lipid MAPS database completely, however, 4 metabolites were not matched. The related content was included as follows: “Eight types of metabolites were suggested as ARA, eicosapentaenoic acid (EPA; C20:5), 11-hydroxy-5Z,8Z,12E,14Z-eicosatetraenoic acid (11-HETE), 12-HETE, 15-HETE, HXB₃, PGG₂, and PGH₂ by comparison with the references in the LIPID metabolites and pathways strategy (MAPS) database (www.lipidmaps.org). However, four types of metabolites did not match the compounds in the Lipid MAPS database. Among them, two types of metabolites as the metabolite numbers 9 and 10 were expected as new hepoxilin and trioxilin analogues, respectively. The other two types of metabolites were not identified because they had many overlapping MS/MS fragments.” (Line 88–97 in the revised manuscript)

(2) We included only critical fragment ions in MS/MS spectra for HXB₃ in Supplementary Fig. 4 in the original manuscript. However, the major fragment ions in the Lipid MAPS structure database looks were also included in our original MS/MS spectra for HXB₃. As the reviewer’s suggestion, all MS/MS spectra were revised by including the major fragment ions in the Lipid MAPS structure with their chemical structures and total molecular masses (Supplementary Fig. 8–10 in the revised manuscript).

The majority of the major fragment ions (i.e., 134, 137, 151, 153, 163, 183, 263, and 289) of HXB₃ in the Lipid MAPS database. The major fragment ions of the compound in the present study were *m/z* 153, 183, 195, and 263 (Supplementary Fig. 9a in the revised manuscript). These fragments can be explained as follows: “The LC-MS/MS fragments of the 12-LOX derived product fragments from ARA showed the peaks at *m/z* 153.2, 183.2, and 195.2 (Supplementary Fig. 9a). The chemical formulas of the two peaks at *m/z* 153.2 and 183.2, which were resulted from the cleavage between C10 and C11 of the HX, were C₁₀H₁₇O and C₉H₁₃OHCOOH, respectively. A peak at *m/z* 195.2 was resulted from the

cleavage between C11 and C12 of the epoxide ring in the HX because the chemical formula was $C_{10}H_{14}OHCOO'$. These fragment peaks indicated that the compound was an HXB₃.” (Line 67–74 Supplementary text in the revised manuscript) The compound was also identified as HXB₃ using NMR analysis as follows: “HXB₃ was identified as (*S*,5*Z*,8*Z*)-10-hydroxy-10-((2*R*,3*S*)-3-((*Z*)-oct-2-en-1-yl)oxiran-2-yl)deca-5,8-dienoic acid (Supplementary Fig. 13). The results of 1D NMR of HXB₃ were shown in Supplementary Table 6. The H-10, H-11, and H-12 had the ROE correlation with each other, indicating the syn geometry (Supplementary Fig. 14a). H-12 was also identified as *S*-form because 12*S*-HpETE was identified as *S*-form. H-5, H-6, H-8, H-9, H-14, and H-15 were confirmed with selective TOCSY irradiation on the peak H-3, H-10, and H-17 (Supplementary Fig. 15). The coupling constants of J₉, J₅, and J₁₅ were ranging below 11 Hz, and H-14 and H-15 showed the ROE correlation, indicating that the double bonds have *Z* geometry (Supplementary Fig. 14b). The 2D NMRs of HXB₃ to support additional structural analysis were shown in the Supplementary Fig. 16.” (Line 146–156 Supplementary text in the revised manuscript)

(3) Thank you for good comment. “Products were compared with the references in LIPID MAPS database” is our wrong expression because HXD₃ and TrXB₃ were novel structures. Thus, these compounds did not match the compounds in the Lipid MAPS database. These points were clarified as the change of ‘HXD₃ and TrXB₃’ to ‘No match (HX analogue) and No match (TrX analogue)’, respectively, in Supplementary Table 3 in the revised manuscript. Also, we revised the text in manuscript as follows: “HxD₃ and TrXB₃ did not match against the compounds in the LIPID MAPS database, suggesting that it is a newly identified metabolite.” was revised to “These metabolites did not match against the compounds in the LIPID MAPS database, suggesting that they are newly identified metabolites.” (Line 94–95 in the revised manuscript). The reviewer #3 commented that substances like PGE₂ and PGH₂ give similar tandem mass spectra and can only be identified taking retention times into account. Thus, the subtitle of ‘Identity’ in the first row was revised to ‘Suggested compounds’ (Supplementary Table 3 in the revised manuscript).

2. Regarding point 1 above, several MS/MS spectra are given for the hepoxilins and trioxilins without reference to the identity of the specific fragment ions. If these have been empirically determined, the diagnostic fragment assignments should be provided. If they have not been determined, this should be stated clearly and some attempt to

assign these fragments should be made.

Answer) MS/MS spectra have been empirically determined. As you checked, the diagnostic fragment assignments and total molecular masses were provided with their chemical structures to assign clearly these fragments (Supplementary Fig. 8–10 in the revised manuscript). More detailed information of MS/MS results were written in Supplementary results sections of “Identification of hydroxy fatty acids, hepoxilins, and trioxilins in the biosynthetic pathways of human lipid mediators by LC-MS/MS and NMR” (Line 22–140 Supplementary text in the revised manuscript).

3. The products were isolated by HPLC by monitoring absorbance at a wavelength of 202 nm. This type of general analysis is not defined for chromophores specific to the proposed structures. Therefore, the chromatogram is not that helpful and should be replaced by something more informative. For example, formation of a conjugated diene in monohydroxy eicosanoids could be monitored at a wavelength of ~235nm, etc...

Answer) Absorbance at 202 nm has been used in HPLC analysis for monitoring non-conjugated HFAs, including HXs and TrXs, and absorbance at 234 nm has been used for monitoring only conjugated HFAs⁶⁵. HXA₃ has been also monitored at 254 nm⁶⁶. In our studies, HXs and TrXs were detected at a wavelength of 202 nm but not at 234 nm and 254 nm. Thus, all products were monitored at 202 nm. (Supplementary Fig. 57 and line 381–385 in the revised manuscript) The related references (#65 and #66) were cited.

4. (1) The authors incubated *M. xanthus* in media containing arachidonic acid for 24 hours and measured the products by LC-MS/MS. They should provide levels of these mediators identified from media alone, as non-enzymatic products could be readily formed under these conditions. (2) The chromatography methods employed would likely not discriminate between stereoisomers and thus it is not clear if the products are racemates. The lack of stereochemically pure standards impedes unequivocal discrimination of enzymatic products from non-enzymatic products. That 11-HETE is one of the major products and is known to be formed non-enzymatically, reinforces this point. (3) Moreover, the authors have named an uncharacterized enzyme based on the presence of 11-HETE (e.g. 11-LOX), which as noted above, can be formed through non-enzymatic oxidation and also via cyclooxygenase (COX) enzymes in low amounts.

Notably, the authors also identified COX homologs in *M. xanthus*. Thus, it is not clear that naming this enzyme “11-LOX” is justified and appropriate at this juncture.

Answer) (1) We already conducted control experiments to check if products are produced by a non-enzymatic reactions, but we did not mentioned it in the original manuscript. As suggested, we incubated the culture medium containing only arachidonic acid without *M. xanthus* for 24 h and measured non-enzymatic products formed under these conditions by HPLC. There were no non-enzymatic products. (Supplementary Fig. 2 and line 83–86 in the revised manuscript)

(2) Only *S*-form of 12-HpETE has been used to convert to HX²⁹. Thus, the chirality of the 12-HETE and 11-HETE products of *M. xanthus* LOXs was determined by chiral phase-HPLC with the pure standards. As a result, the products were identified as *S*-forms. (Supplementary Fig. 12 and 164–167 in the revised manuscript) The related references (#29) were cited.

(3) The gene encoding the putative LOX from *M. xanthus* was cloned and expressed in *E. coli* BL21 (Line 310–319 in the original manuscript). The putative LOX from culture broth was purified (Line 321–331 in the original manuscript), and the substrate specificity of the purified enzyme for polyunsaturated fatty acids was measured (Line 333–339 in the original manuscript). The enzyme exhibited the highest activity for arachidonic acid and its product was 11-HpETE, which was reduced to 11-HETE (Supplementary Table 5 in the original manuscript). The compound was identified as 11-HETE by the same retention time in HPLC as the standard 11-HETE. Thus, the putative LOX from *M. xanthus* was identified as a 11-LOX.

5. (1) Following from above, to my knowledge, there is no mammalian homolog of “11-LOX” and thus it is unclear if the products formed (i.e., the novel HXD₃ and TrXD₃) from this bacterial culture would be present or (2) of physiological relevance in human tissue. To this end, for novel structures encountered from the bacteria, the authors should determine whether these products are present in mammalian tissues. This analysis would be aided by the biogenic standards they have isolated and could be significantly enhanced by using deuterium-labeled precursors (ie., arachidonic acid) to generate labeled products for use as internal standards.

Answer) (1) It is very clear that the novel HXD₃ and TrXD₃ were produced by the purified 11-LOX and epoxide hydrolase (EH). (Supplementary Table 5 in the revised manuscript)

HXD₃ and TrXD₃ were also identified from culture of recombinant *E. coli* expressing 11-LOX and EH from *M. xanthus* by HPLC, LC-MS/MS, and NMR. (Fig. 2, Supplementary Fig. 9, 10, 25, 41, and Supplementary Table 5 in the revised manuscript)

(2) To the best of our knowledge, the presence and physiological relevance of the bacterial 11-LOX-derived products such as HXD₃ and TrXD₃ in human tissue have not been reported. The experiments for the presence or absence of these products in mammalian tissues using ¹⁴C-labeled arachidonic acid were performed by other research groups^{6,27}. As a result, HXD₃ and TrXD₃ were not found in human and rat tissues. Thus, not only 11-LOX but also 11-LOX-derived products such as HXD₃ and TrXD₃ do not exist in mammalian tissues. (Line 236–242 in the revised manuscript) The related references (#6 and #27) were cited.

6. (1) The authors used preparative HPLC to purify biogenic compounds and subsequently performed 2D-NMR to determine their structures. However, very little information is given in the text regarding the structures. (2) Moreover, there is no mention of the stereochemistry given in the text and whether stereochemically pure products were obtained.

Answer) (1) As suggested, the detailed information regarding the structures by determining NMR was newly included in line 141–246 of the Supplementary text in the revised Supplementary manuscript. Moreover, the supporting data were added and arranged (Supplementary Table 6–14 and Supplementary Fig. 13–48).

(2) We purified products with high purity using Prep-HPLC (Supplementary Fig. 11). We confirmed and added the results of stereoselectivity for the intermediates 12*S*-HETE and 11*S*-HETE and newly included in the text as follows: “Only *S*-form of 12-HpETE has been used to convert to HX in nature²⁹. Thus, the chirality of the 12-HETE and 11-HETE products of *M. xanthus* LOXs was determined by chiral phase-HPLC with the pure standards. As a result, the products were identified as *S*-forms.” (Supplementary Fig. 12 and 164–167 in the revised manuscript) The related references (#29) were cited. 11*S*-HpETE and 12*S*-HpETE with high purity were used as standards for analyzing all HXs and TrXs. The chiral centers of HXs were obtained from the *S*-form HpETEs with their fixed chiral centers, and the chiral centers of TrXs were determined by those of HXs because TrXs were obtained from HXs. HXs and TrXs were diastereomer, so the stereochemistry was confirmed by ROESY NMR. Therefore, their results were newly included in line 141–246 of the Supplementary text in the revised

Supplementary manuscript.

7. (1) Regarding the conversion of substrate to product in *E. coli* expressing the *M. xanthus*-derived enzymes, a comparative quantification of the hepoxilins and trioxilins with and without the presence of epoxide hydrolase (EH) should be shown in a combined graph. (2) At present, hepoxilin levels are only provided in *E. coli* incubations without the presence of EH and trioxilin production is only quantified in bacteria with EH. Was conversion 100% complete? To understand these product-precursor relationships, all compounds should be measured and presented for each incubation.

Answer) (1) As suggested, a comparative quantification of the hepoxilins and trioxilins with and without the presence of epoxide hydrolase (EH) was shown in a combined graph (Fig. 2 in the revised manuscript).

(2) All compounds, including polyunsaturated fatty acids (PUFAs), hydroperoxy fatty acids (HPFAs), hydroxy fatty acids (HFAs), hepoxilins (HXs), and trioxilins (TrXs), were measured and presented for each incubation (Fig. 2 in the revised manuscript). Moreover, the conversion of HX to TrX was not 100% complete. Therefore, trioxilin levels were newly included in *E. coli* incubations without the presence of EH (Fig. 2a,b,c,d and Supplementary Fig. 49 in the revised manuscript).

8. Following from above, the authors never provide levels of any of the lipid mediators in *E. coli* in absence of expression of the enzymes derived from *M. xanthus*, making it difficult to connect biosynthesis solely to expression of the enzymes (again, some of these products could be formed non-enzymatically during the incubations with *E. coli*).

Answer) We already conducted control experiments to check if products are produced by a non-enzymatic reactions, but we did not mentioned it in the original manuscript. As suggested, the time-course reactions using *E. coli* in absence of expression of the enzymes derived from *M. xanthus* were performed under the same conditions of 50 mM EPPS (pH 8.5) buffer containing 1 mM arachidonic acid and 7.2 g L^{-1} cells at 30°C for 120 min, and measured non-enzymatic products by HPLC. However, there were no non-enzymatic products (Supplementary Fig. 7 in the revised manuscript).

9. With the *E. coli* or isolated enzyme incubations (e.g., Figs 2 and 3), it is not clear from

the methods how the products were quantified at each time point. Was an aliquot collected and subjected to LC-MS/MS at each time point? How was the accurate concentration of each product determined? Were molar extinction coefficients empirically determined for the new products?

Answer) The concentrations of polyunsaturated fatty acids (PUFAs), hydroperoxy fatty acids, hydroxy fatty acids, hepxilins, and trioxilins were determined using an HPLC system using an HPLC system (Agilent 1260) with a reversed-phase Nucleosil C18 column (3.2×150 mm, 5- μ m particle size; Phenomenex). The PUFA standards ARA, EPA, and DHA, and the hydroxy fatty acid (HFA) standards 11-HETE, 12-HETE, 12-HEPE, and 14-HDoHE were purchased from Sigma (St. Louis, MO, USA) and Cayman Chemical (Ann Arbor, MI, USA), respectively. The hepxilin and trioxilin standards were prepared as described in the section of 'Materials' (Line 281–296 in the original manuscript). The reaction products were identified to have the same retention times as those of their corresponding standards. The amounts of the products were calculated by calibrating the peak areas to the concentrations of the polyunsaturated fatty acid, hydroperoxy fatty acid, hydroxy fatty acid, hepxilin, and trioxilin standards using molar extinction coefficients. (Line 390–392 in the revised manuscript) In detail explanation was follows: First, the standards of all compounds are prepared at concentrations of 5-points between 0.1 and 5 mM. Second, the area value of the standards at each concentration were measured using HPLC (Supplementary Fig. 58a). Third, the areas were assigned to each concentration to obtain a calibration curve (Supplementary Fig. 58b). Finally, the concentrations of products were determined from the peak areas using the calibration curve. For an example, a calibration curve for the peak areas to the concentrations of the HXB₃ standard using molar extinction coefficients and a graph with the peaks at the several concentrations of the HXB₃ standard were presented in Supplementary Fig. 58 in the revised manuscript.

The hepxilin and trioxilin standards were prepared by Prep-HPLC. To show the purity of these standards, the HPLC profiles before and after Prep-HPLC were newly included. The collected samples showed >99% purity (Supplementary Fig. 11 and line 302–314 in the revised manuscript). Additionally, the molecular mass of the new products as HXD₃ was determined by calculation of the molecular chemical formula.

10. (1) The authors present results suggestive of agonist activity of the hepxilins and

trioxilins for PPAR γ . However, high micromolar concentrations were used for these assays and it is not clear how these concentrations were justified (see Point 9 above). (2) Moreover, there is no statistical analysis presented to determine whether the increase in activity is significant or not. Overall, the PPAR γ activity assay seems to be completely out of place. (3) The physiological relevance of this is not at all clear. Given the documented pro-inflammatory roles of hepoxilins, such as promoting neutrophil transmigration, vascular permeability, and hyperalgesia, it is not clear why activation of PPAR γ has been selected as the sole readout of the potential biological activity of these products. (4) The suggestion that these new products “can be developed as new generation drugs instead of conventional agonists, TMZ’s” ..”because of known side effects”, is a highly speculative statement and should be removed.

Answer (1) The concentrations of hepoxilins and trioxilins used for PPAR γ assay were justified as described above (see Answer #9). In other reports, the concentrations of partial agonists were used as high micromolar concentrations ranging from 5 to 20 μ M. Therefore, we used in the range of 10 to 20 μ M lipid mediator in the present study. The sentence of “The concentrations of TRO and lipid mediators were the same as those used in other reports^{50, 64}.” was newly added. (Line 367–368 in the revised manuscript) The references (#50 and #64) were newly cited.

(2) We performed statistical analyses as follows: “Statistical analyses. The means and standard errors for all experiments were quantitatively calculated with one-way analysis of variance (ANOVA) from triplicate experiments. ANOVA was carried out using Tukey’s method, with a significance level of a P value of 0.05 using SigmaPlot 10.0 (Systat Software, Chicago, IL, USA).” (Line 396–399 in the revised manuscript). All figures in Fig. 3 and Supplementary Fig. 51 were revised by the use of statistical analyses.

(3) The activation of PPAR γ was selected as the potential biological activity of these lipid mediators because “PPAR γ is an important nuclear receptor closely related to biological activities as lipid accumulation^{46, 47}, glucose metabolism⁴⁸, the inflammatory response^{49, 50}, neutrophil transmigration⁵¹, vascular permeability¹⁸, and hyperalgesia^{52, 53}.”(Line 258–260 in the revised manuscript) The related references (#46–#53) were cited.

(4) As suggested, the sentence of “can be developed as new generation drugs instead of conventional agonists, TMZ’s” was removed. To express more clearly, the sentence of “The identification of natural PPAR γ agonists has been required because of the harmful side effects

of TMZs, antidiabetic and anti-inflammatory drugs.” were replaced to “The identification of natural partial agonists of PPAR γ has been required for development of the antidiabetic and anti-inflammatory drugs with reduced side effects.” (Line 293–294 in the revised manuscript). Partial agonists showed reduced side effects. The related contents were included as follows: “Thiazolidinediones (TMZs) such as rosiglitazone⁵⁴ and TRO⁵⁵ are representative agonists of PPAR γ as full agonists. Full agonists have side effects such as weight gain and worsening of congestive heart failure^{56,57}, whereas partial agonists retain beneficial anti-diabetic properties with reduced side effects²⁹. Partial agonists are defined as weak activators of PPAR γ that show the same activation pattern with lower transactivation potential compared to full agonists⁵⁸.” (Line 260–265 in revised manuscript) The related references (#57–#58) were newly cited.

11. Results are presented of incubations with isolated purified enzymes (i.e., 12-LOX and so-called 11-LOX of bacterial origin). However, no information is given regarding the purity of the enzyme preparations.

Answer) Thank you for good comment. To show the purity of all candidate enzymes originated from *M. xanthus*, SDS-PAGE analysis was newly included (Supplementary Fig. 4 in the revised manuscript). Moreover, information was given regarding the purity of the enzyme preparations as follows: “The recombinant 12-LOX and 11-LOX enzymes were purified from a crude cell extracts as single soluble proteins using His-Trap affinity chromatography (Supplementary Fig. 4a) with specific activities of 605 and 487 $\mu\text{mol min}^{-1} \text{mg}^{-1}$, respectively.” (Line 115–117 in the revised manuscript)

12. For the comparative sequence analysis, the authors should provide information about the alignment with the human enzymes and % sequence similarity.

Answer) Thank you for good comment. As suggested, the information about the alignment with the human enzymes and % sequence similarity was provided as follows: “Although the amino acid sequences of these enzymes showed 15–40% identities with human corresponding enzymes, the major residues affecting the activity were conserved (Supplementary Fig. 3).” (Line 106–108 in the revised manuscript). The figure of sequence alignment was newly included (Supplementary Fig. 3). Moreover, the column of ‘identity (%)’ was newly added, and identities with human corresponding enzymes were newly included in

Supplementary Table 4 in the revised manuscript.

Response to Reviewer #2

1. There is no statistical analysis of the results presented in Figure 4.

Answer) It is our mistakes. As suggested, we included newly statistical analyses as follows: “Statistical analyses. The means and standard errors for all experiments were quantitatively calculated with one-way analysis of variance (ANOVA) from triplicate experiments. ANOVA was carried out using Tukey’s method, with a significance level of a P value of 0.05 using SigmaPlot 10.0 (Systat Software, Chicago, IL, USA).” (Line 396–399 in the revised manuscript). All figures in Fig. 3 and Supplementary Fig. 51 were revised by the use of statistical analyses.

2. What is the evidence that the compounds tested are not metabolized by the human embryonic kidney (HEK) cells? Metabolites of the compounds may be the active substances.

Answer) “Human embryonic kidney (HEK) 293 cells have been widely used for the screening of PPAR γ agonist⁶³, suggesting that the endogenous factor of HEK293 cells do not affect the studies of PPAR γ agonists. The lipid mediators tested were not metabolized by HEK 293 cells (Supplementary Fig. 56).” (Line 357–360 in the revised manuscript) The metabolites were investigated by HPLC analysis after cultivation of the HEK cells with lipid mediator transfected with plasmids for 24 h. (Supplementary Fig. 56 in the revised manuscript). The reference (#63) was newly cited.

3. (1) Are the compounds that increase the transcriptional activity of PPAR γ bona fide PPAR γ ligands? An *in silico* docking analysis could have been done which show (or not) that the compounds can fit into ligand binding pocket of PPAR γ . (2) In addition, although technically more demanding, would be *in vitro* experiments testing the ability of the active compounds to recruit a co-activator to the ligand binding domain.

Answer) (1) “The PPAR γ ligand binding domain (LBD) consists of a bundle of 13 α -helices and 4 β -strands, and agonists are bind to helices H3, H5, H10, and H12, including the major residues Ser289 (H3), His323 (H5), His449 (H10), and Tyr473 (H12)⁵⁹.” (Line

266–268 in the revised manuscript) The compounds that increase the transcriptional activity of PPAR γ were HXB₃, HXB₄, HXD₃, TrXB₃, TrXD₃, and 11-HETE. As you suggested, we carried out *in silico* docking analysis as follows: “These compounds were docked to LBD at PPAR γ using molecular models (Supplementary Fig. 54). Rosiglitazone, known as a full agonist, was interacted with Tyr473 (H12) (Supplementary Fig. 55a), whereas HXB₃, HXB₄, HXD₃, TrXB₃, TrXD₃, and 11-HETE were interacted with Ser289 (H3) and His323 (H5) (Supplementary Fig. 55b,c,d,e,f,g). The docking results suggest that HXB₃, HXB₄, HXD₃, TrXB₃, TrXD₃, and 11-HETE are partial agonists.” (Line 275–280 in the revised manuscript).

(2) In the present study, we could not perform *in vitro* experiments testing the ability of the active compounds to recruit a co-activator to the ligand binding domain because the experiments require many time and effort and can be considered as another manuscript. Please understand this situation.

4. Figure 4. Troglitazone increases the transcriptional activity of PPAR γ by 2.5 to 3.5 fold. In Figure 4b, the activation is very weak indicating that this experiment has not worked well, but lead to the conclusion that HXB₄ is not an activator. If the control has not worked well, this conclusion may be wrong.

Answer) Thank you for your careful checking. As suggested, the activation of troglitazone was very weak indicating that this experiment had not worked well. Therefore, we tried again the experiment carefully. In the confirmed experiment, troglitazone increases the transcriptional activity of PPAR γ by 3-fold, indicating that this experiment was reliable. In newly data, The degree of up-regulation of HXB₄ on the transcriptional activity of PPAR γ was similar to that by TRO. Thus, the sentence of “HXB₄ and HXB₅ (TrXB₄, and TrXB₅) did not affect the PPAR γ activity (Fig. 3b,c,f,g). When TRO was supplemented, HXB₄ up-regulated PPAR γ activity slightly, whereas HXB₅ down-regulated it.” were replaced to “The degree of up-regulation of HXB₄ on the transcriptional activity of PPAR γ was similar to that by TRO (Fig. 3b). HXB₅, TrXB₄, and TrXB₅ did not affect the PPAR γ activity (Fig. 3c,f,g). When TRO was supplemented, HXB₅ down-regulated PPAR γ activity. 11-HETE and 12-HETE as intermediate products showed effects similar to those of HXB₃ and HXB₅, respectively, on PPAR γ activity (Supplementary Fig. 51).” (Line 203–207 in the revised manuscript).

5. (1) For active compounds (HXB₃, HXD₃, TrXB₃, TrXD₃) and additive effect if observed when they are tested together with troglitazone. How is this being explained? (2) Have experiments been done with lower or higher concentrations than 3 μM of troglitazone?

Answer) (1) The additive effect if observed when they are tested together with troglitazone can be explained as follows: “Full agonists of PPAR γ such as TMZs are known to bind to H12, whereas partial agonists stabilize the β -sheet and the H2'/H3 area, resulting in distinct transcriptional effects between full and partial agonists⁶⁰. Thus, the synergetic effects of full and partial PPAR γ agonists exist.” (Line 268–271 in the revised manuscript) The related references (#60) were newly cited.

(2) In other reports, 3 μM troglitazone has been used with partial agonists and the concentrations of partial agonists were ranging from 10 to 20 μM. Therefore, we used 3 μM troglitazone and less than 20 μM lipid mediator in the present study. The sentence of “The concentrations of TRO and lipid mediators were the same as those used in other reports^{50,64}.” was newly added. (Line 367–368 in the revised manuscript) The references (#50 and #64) were newly cited.

6. The highest concentration tested for compounds is 20 μM. For active compounds there is no evidence that the maximum effect is reached at this concentration.

Answer) We agree the reviewer’s claim that the maximum effect is reached at this concentration. The physiological treatment at too high concentrations of active compounds was meaningless, and it showed cytotoxicity. Thus, we used up to 20 μM as an arbitrary concentration in the experiments.

Response to Reviewer #3

1. (1) The authors incubated ARA for 24 hours with the bacteria. How did the authors control for autoxidation phenomena? Why is no data of control incubations presented? Did the authors investigate the employed ARA batch for the presence of autoxidation products? (2) As very high amounts of PUFA were used it is mandatory to test the starting materials for impurities, autoxidation signals, has this been tested?

Answer) (1) Thank you for your careful checking. We incubated *M. xanthus* in medium

containing arachidonic acid for 24 h (Line 81–86 in the original manuscript). As suggested, we tested the starting materials for impurities and autoxidation signals. We incubated only culture medium containing arachidonic acid without wild *M. xanthus* for 24 h and measured autoxidation phenomena by HPLC. However, there was no autoxidation. (Supplementary Fig. 2 in the revised manuscript)

The related content was newly included as follows: “*M. xanthus* consumed most of ARA and the peaks of some metabolites were detected (Supplementary Fig. 2d), whereas ARA was not consumed and not detected any new peaks in the culture medium without *M. xanthus* (Supplementary Fig. 2c).” (Line 83–86 in the revised manuscript)

(2) The autoxidation signals of PUFA were measured as described above (1). The impurities of PUFA were measured by HPLC, impurities showed little as shown in Supplementary Fig. 2b in the revised manuscript.

2. (1) The presented MS/MS spectra seem very clean to me, why is there basically no noise? (2) Was this such high concentrations used to obtain these data? Was this data obtained from pure standards or incubations?

Answer) (1) The presented MS/MS spectra was clean because only critical fragments were presented by removing noise. To present more exactly MS/MS spectra, we changed the treated MS/MS spectra to the original MS/MS spectra with noise. (Supplementary Fig. 8–10 in the revised manuscript)

(2) The exact concentrations to obtain MS/MS spectra were unknown because these data were obtained from incubations.

3. (1) The NMR data which apparently was used for the accurate structural elucidation seems inappropriate. The authors specify double bond geometry as well as stereochemistry. I am wondering how the authors could define absolute stereochemistries? (2) Using NMR one would need to label with Mosher's acid chloride or use shift reagents, this is not apparent from the manuscript. (3) Alternatively chiral chromatography and comparison with standards would also suffice or the use of circular dichroism, again, maybe I missed it but this is not obvious from the text. (4) Next, the authors give double bond geometries, however the NMR tables in the supplement do not contain the coupling constants for the double bond protons, which

would be important to judge the *E/Z* geometry. (5) Alternatively I would have seen DQF-COSY or Jres and NOESY as possible methods to specify the coupling constants and the geometry. The authors give a ROESY spectrum image which actually looks like a NOESY but without any information about peak integrals (or intensities) that could assist the analysis of stereochemistry. Furthermore, this spectrum image clearly contains TOCSY peaks too, and thus, it is not clear how the authors determined the stereochemistry using this data.

Answer) (1) Thank you for good comment. 11-HpETE and 12-HpETE with high purity were identified to be *S*-configuration using CP-HPLC, and thus stereochemistry analysis for all of HXs and TrXs was performed based on *S*-form configuration of HpETEs. HXs were obtained from *S*-form HpETEs and their chiral centers did not change. The chiral centers of HXs were obtained from the *S*-form HpETEs with their fixed chiral centers, and the chiral centers of TrXs were determined by those of HXs because TrXs were obtained from HXs. HXs and TrXs were diastereomer, so the stereochemistry was confirmed by ROESY NMR. These results were newly included in line 141–246 of the Supplementary text in the revised manuscript.

(2) As mentioned by the reviewer, we tried to treat a chiral shift reagent (CSR), Europium(III) tris[3-(heptafluoropropylhydroxymethylene)-*d*-camphorate], and expected to get the stereochemistry information and also separate the overlapped peaks of compounds including double bonds. However, when CSR was added to HXs and TrXs, all of peaks became broad and the peaks of chiral centers disappeared. (1.2–3 mM of CSR was added to 10 mM of HXs and TrXs) The chiral centers composed of alcohol or epoxide seemed to be affected by CSR binding. The result was as follows.

Figure. ^1H NMR of (a) CSR. (b) TrXB_5 + CSR. (c) TrXB_5 . (d) HXA_4 + CSR. (e) HXA_4 . (f) HXD_3 + CSR. (g) HXD_3 .

(3) There are no standards of HXs and TrXs up to the present. Instead, we purified 11-HETE and 12-HETE with high purity as products using Prep-HPLC (Supplementary Fig. 11). We also confirmed and added the results of stereoselectivity for the intermediates 12*S*-HETE and 11*S*-HETE and newly included in the text as follows: “Only *S*-form of 12-HpETE has been used to convert to HX in nature²⁹. Thus, the chirality of the 12-HETE and 11-HETE products of *M. xanthus* LOXs was determined by chiral phase-HPLC with the pure standards.

As a result, the products were identified as *S*-forms.” (Supplementary Fig. 12 and line 163–167 in the revised manuscript) The related references (#29) were cited. 11*S*-HpETE and 12*S*-HpETE with high purity were used as standards for analyzing all HXs and TrXs. The chiral centers of HXs were obtained from the *S*-form HpETEs with their fixed chiral centers, and the chiral centers of TrXs were determined by those of HXs because TrXs were obtained from HXs. HXs and TrXs were diastereomer, so the stereochemistry was confirmed by ROESY NMR.

(4) By selective-TOCSY NMR, the overlapped peaks of double bonds of HXs and TrXs were effectively separated and the approximate coupling constants of double bonds were obtained to their *E/Z* geometry information. ROESY NMR was also used to decide *E/Z* geometry of double bonds. However, we could not get the exact coupling constants since the double bond peaks were much overlapped of 6~8 protons.

(5) Thank you for your good comments. We recognized the lack of explanation for the structural analysis using NMR in the original manuscript. Thus, we described the results of additional analysis of ROSEY and selective-TOCSY NMR as described above, and added text and figures. (Supplementary Fig. 13–48 and line 141–246 Supplementary text in the revised manuscript).

4. The authors also state to have identified lipid mediators by comparison with Lipidmaps. This is inappropriate. Substances like PGE₂ and PGH₂ give similar tandem mass spectra and can only be identified taking retention times into account. Again it is not obvious to me that this has been done.

Answer) Thank you for your good comment. We agree that substances like PGE₂ and PGH₂ give similar tandem mass spectra and can only be identified taking retention times into account. Thus, the sentence of “COX from *M. xanthus* converted ARA to PGH₂, and its activity toward ARA was 0.011 μmol min⁻¹ mg⁻¹ (Supplementary Fig. 5).” was revised to “COX from *M. xanthus* converted ARA to PG analogues, and its activity toward ARA was 0.011 μmol min⁻¹ mg⁻¹ (Supplementary Fig. 5).” (Line 111–112 in the revised manuscript). Also, the sentence of “The metabolite was identified as PGH₂.” was revised to “The metabolite was suggested as PGH₂.” (Legend of Supplementary Fig. 2 in the revised manuscript).

Reviewers' comments:

Reviewer #1 (Remarks to the Author):

The authors were responsive to my previous comments in their revised manuscript submission. However, as detailed below, there are several issues with the manuscript that remain in its current form.

Introduction:

Several imprecise statements are included and should be revised or removed. For example, the statement (line 46) “high-level production and more diverse types of HXs and TrXs are required” is unclear. What is “high-level” and why are “more diverse types” required?

The statement (Line 47) “LOXs are the starting enzymes for the biosynthesis of human lipid mediators” should be revised. In addition to LOXs, several other enzymes participate in lipid mediator biosynthesis, including the cytochrome P450 families, cyclooxygenases, and others.

The use of the term “Human lipid mediators” in the title and in several places in the manuscript is unnecessary. Lipid mediators have conserved structures. Given that the new products have not even been identified in human tissues, it is unclear why this term has been used in the title and throughout the manuscript.

Line 71, “Until now, human lipid mediators have never been found in bacteria” is misleading. Previous studies have found microbial (e.g., *Bacillus megaterium*) isoforms of cytochrome P450 enzymes that actively convert polyunsaturated fatty acids to hydroxylated products of arachidonic acid and eicosapentaenoic acid that are also produced in human cells and tissues. Moreover, other studies have defined microbial (e.g., *Pseudomonas aeruginosa*) lipoxygenases and epoxide hydrolases that are biologically active.

The statement (Line 76) “the transcriptional activity of PPAR γ for HXs and TrXs was determined to investigate their antidiabetic and anti-inflammatory activities” should be removed. The authors did not investigate antidiabetic or anti-inflammatory activities of these products.

Results:

Line 94, "These metabolites did not match against the compounds in the LIPID MAPS database, suggesting that they are newly identified metabolites" should be revised. The LIPID MAPS database does not contain all lipid mediators known in the literature and thus absence of a lipid product in the database does not necessarily imply that it has not been described before.

Line 107, "Although the amino acid sequences of these enzymes showed 15-40% identities with human corresponding enzymes, the major residues affecting the activity were conserved". No references are given to support the designation of specific residues as substrate binding and so forth.

Line 115, it is unclear why the authors suddenly call one of the enzymes (MXAN_1744) an 11-LOX at this point in the manuscript when this gene and the other (MXAN_1745) simply have some sequence homology with mammalian lipoxygenases. At this point in the manuscript, it has not been defined which enzyme produces 11-HETE.

Discussion:

Line 275, states "...demonstrated PPAR γ up-regulation". Only luciferase activity was reported, while expression of PPAR γ was not evaluated.

In general, the results with PPAR γ activation are overstated. The authors have simply shown increased activity of the luciferase reporter, as has been demonstrated for numerous other exogenous compounds. This study does not demonstrate direct agonist activity, despite the in silico docking studies. Secondary production of other endogenous ligands in these cells cannot be ruled out. Also, in response to my previous comment regarding justification of the concentration used to test for PPAR agonist activity, the authors simply state that "In other reports...high micromolar concentrations were used". This is not a scientific justification for these particular products.

Reviewer #2 (Remarks to the Author):

This revised manuscript is of higher quality than the first submitted version. However, I would like to come back to one point which relates to lines 268-271 of the revised manuscript. The authors write "Full agonists of PPAR γ such as TMZs are known to bind to H12, whereas partial agonists stabilize the β -sheet and the H2'/H3 area, resulting in distinct transcriptional effects between full and partial agonists. Thus, the synergetic effects of full and partial PPAR γ agonists exist." It is not clear what is meant here. If the mentioned synergistic effect as been observed by

others a ref should be added. Unless it has escaped my attention, there is no synergistic effect shown in this paper, at best some additive effects (see results of Fig. 3).

Reviewer #3 (Remarks to the Author):

Martin Giera

The authors have in general responded well. However, some questions remain as outlined below.

Title: the title might sound a little bit more concise if it would just read: "Biotransformation of polyunsaturated fatty acids to bioactive hepoxilins and trioxilins by microbial enzymes"

Considering the earlier points raised by myself:

1. Ok, thank you for the clarification.
2. I have to admit that I personally do find it rather uncommon to remove signals from obtained spectra, stating that only critical fragments remained. In other words, how am I supposed to judge the presented spectra if I do not exactly know how the actual spectra looked like and I only get to see pre-treated data? Ok, good that this is now to be found in the supplement.
3. The authors rely heavily on the presence of ROESY crosspeaks for determining whether a pair of protons attached to adjacent carbons is in the E or Z (or syn or anti) configuration. I doubt whether this is by itself sufficient, because fig 14a in the suppl. info also shows a crosspeak between H10 and H7, protons that are presumably further separated than H14 and H15 in a hypothetical E configuration. Perhaps the authors can give a reference that demonstrates that ROESY is a suitable method for this purpose. And might coupling constant not also actually help to decipher the actual geometries?
4. The authors actually agree with me that only when taking retention times into account PGs can be identified. Still authors do not carry out such experiment and instead state the metabolite is suggested to be... I do not understand why such a simple experiment is not carried out. It takes half a day to inject authentic standards along with an internal standard and compare relative retention times. At least PGE2 and PGD2 and several other PGs are readily available.

Manuscript p5 l94. The sentence..."These metabolites did not match against the compounds in the LIPID MAPS database, suggesting that they are newly identified metabolites" does not seem entirely correct to me. In other words, any other sources than LIPID MAPS are left out of consideration, however if a lipid is not in LIPID MAPS doesn't mean i) it has not yet been identified and ii) it has not been published elsewhere.

Maybe rephrase?

Responses to Reviewers:

Response to Reviewer #1

1. (1) Several imprecise statements are included and should be revised or removed. (2) For example, the statement (line 46) “high-level production and more diverse types of HXs and TrXs are required” is unclear. (3) What is “high-level” and why are “more diverse types” required?

Answer) (1) Thank you for your good comment. As you suggested, several imprecise statements were revised as follows: The term of “peroxisome proliferator-activated receptor- γ partial agonists” was revised to “potential peroxisome proliferator-activated receptor- γ partial agonists”. **(Line 17 in the second revised manuscript)** The term of “will” in “These findings will facilitate physiological studies and drug development based on lipid mediators” was revised to “may”. **(Line 18 in the second revised manuscript)** The term of “partial agonists” was revised to “potential partial agonists”. **(Line 297 in the second revised manuscript)** The term of “will” in “Our achievement provides practical advances in the field of lipid mediators and will stimulate physiological studies and drug development on lipid mediators” was revised to “may”. **(Line 300 in the second revised manuscript)** The sentence “However, other PGs converted from PGH₂ were not found in the bacterium.” **(Line 144–145 in the**

revised manuscript) was removed because of inexact result. The title in Results “Identification of ARA-derived metabolites of *M. xanthus* by LC-MS analysis” revised to “LC-MS analysis for ARA-derived metabolites of *M. xanthus*” **(Line 82 in the revised manuscript)** The term of “newly identified metabolites” was revised to “new metabolites”. **(Line 98 in the second revised manuscript)** The term of “identified” was revised to “suggested”. **(Line 99 in the second revised manuscript)**

The other several imprecise statements were revised or removed as shown in the following answers.

(2) As suggested, the related content for the statement **(Line 46 in the revised manuscript)** was revised to describe more clearly and exactly the content as follows: The sentences of “Thus, HXs and TrXs are important lipid mediators. For the development of drugs based on these compounds, the high-level production and more diverse types of HXs and TrXs are required.” was changed to “Thus, HXs and TrXs are important lipid mediators. However, the produced concentrations of HXs and TrXs are too low and their types are limited to less than 10. For the discovery of bioactive compounds as potential drugs, more efficient production and diverse types of HXs and TrXs are required.”. **(Line 44–47 in the second revised manuscript)**

(3) The term of “high-level” was revised to “efficient”. The reason for requirement of “more diverse types” is the discovery of bioactive compounds as potential drugs.

2. The statement (Line 47) “LOXs are the starting enzymes for the biosynthesis of human lipid mediators” should be revised. In addition to LOXs, several other enzymes participate in lipid mediator biosynthesis, including the cytochrome P450 families, cyclooxygenases, and others.

Answer) As suggested, in the sentence, the term of “LOXs” was revised to “LOXs, cyclooxygenases (COXs), and the cytochrome P450 families”, and “Human” was removed **(Line 48–49 in the second revised manuscript)**. The sentence of “LOXs, a family of non-heme-iron-containing dioxygenases” was revised to “Among these enzymes, LOXs, a family of non-heme-iron-containing dioxygenases”. **(Line 49–50 in the second revised manuscript)**

3. The use of the term “Human lipid mediators” in the title and in several places in the manuscript is unnecessary. Lipid mediators have conserved structures. Given that the

new products have not even been identified in human tissues, it is unclear why this term has been used in the title and throughout the manuscript.

Answer) Thank you for your good comment. The term of “human” in “human lipid mediators” was absolutely removed for all contents of the manuscript and supplementary information through four different revision.

- 1) The term of “human lipid mediators” was removed. **(Line 2, 7, 102, 157, and 193 in the revised manuscript)**
- 2) The term of “human lipid mediators” was revised to “lipid mediators”. **(Line 19, 49, 61, 128, 257, 273, 294, 295, 296, and 300 in the second revised manuscript)**
- 3) The term of “human lipid mediators” was specified to “HXs and TrXs”, “HXs, TrXs, and PGs”, or “HETE, HXs, and TrXs”. **(Line 11, 73, 74, 75, 100, 131, 180, 181, 197, 223, 247, 288, 289, and 291 in the second revised manuscript)**
- 4) The term of “human lipid mediators” was revised to “the lipid mediators”, which indicated LTs, LXs, RVs, PTs, PGs, HXs, and TrXs. **(Line 62, 64, and 72 in the second revised manuscript)**

4. Line 71, “Until now, human lipid mediators have never been found in bacteria” is misleading. Previous studies have found microbial (e.g., *Bacillus megaterium*) isoforms of cytochrome P450 enzymes that actively convert polyunsaturated fatty acids to hydroxylated products of arachidonic acid and eicosapentaenoic acid that are also produced in human cells and tissues. Moreover, other studies have defined microbial (e.g., *Pseudomonas aeruginosa*) lipoxygenases and epoxide hydrolases that are biologically active.

Answer) Thank you for your careful checking. Hydroxy-eicosanoids, which were converted by microbial enzymes were discovered already, and several related enzymes were defined. But, HXs, LTs, and PGs, which have 5-ring or epoxide group, and TrXs, were not reported in bacteria yet. Although some microbial lipoxygenases were reported, the microbial activities for epoxidation were not discovered yet. Thus, the term of “human lipid mediators” was replaced more specific “HXs, TrXs, and PGs” in the sentence. **(Line 73 in the second revised manuscript)**

5. The statement (Line 76) “the transcriptional activity of PPARy for HXs and TrXs was

determined to investigate their antidiabetic and anti-inflammatory activities” should be removed. The authors did not investigate antidiabetic or anti-inflammatory activities of these products.

Answer) We agree the reviewer’s comment. In this study, only transcriptional activity of PPAR γ for HXs and TrXs was reported. Thus, the sentence of “to investigate their antidiabetic and anti-inflammatory activities” was removed. **(Line 78 in the revised manuscript)**

6. Line 94, “These metabolites did not match against the compounds in the LIPID MAPS database, suggesting that they are newly identified metabolites” should be revised. The LIPID MAPS database does not contain all lipid mediators known in the literature and thus absence of a lipid product in the database does not necessarily imply that it has not been described before.

Answer) Thank you for your good check. As suggested, we compared new compounds in several other databases, including PubChem, the Human Metabolome Database, and KEGG, and research papers besides LIPID MAPS database. Nevertheless, we did not found the same compounds as the metabolite numbers 9 and 10. Thus, the sentence of “These metabolites did not match against the compounds in the LIPID MAPS database, suggesting that they are newly identified metabolites.” was revised to “These metabolites did not found against the compounds in information databases, including the LIPID MAPS Database, PubChem, the Human Metabolome Database, and KEGG, and their MS/MS profiles have not been reported elsewhere to date. Thus, they are suggested as new metabolites. ”. **(Line 95–98 in the second revised manuscript)**

7. Line 107, “Although the amino acid sequences of these enzymes showed 15-40% identities with human corresponding enzymes, the major residues affecting the activity were conserved”. No references are given to support the designation of specific residues as substrate binding and so forth.

Answer) As suggested, several references were given to support the designation of specific residues as substrate binding and so forth. We used human enzymes as templates for sequence alignment, and the structure studies of human enzymes were reported already. To explain the designation of specific residues, we added references as follows: “Although the amino acid

sequences of these enzymes showed 15–40% identities with human corresponding enzymes^{23, 24, 25, 26, 27, 28, 29}, the major residues affecting the activity were conserved (Supplementary Fig. 3).” **(Line 108–110 in the second revised manuscript)** The references **(#23, #24, #25, #26, #27, #28, and #29)** were newly cited.

8. Line 115, it is unclear why the authors suddenly call one of the enzymes (MXAN_1744) an 11-LOX at this point in the manuscript when this gene and the other (MXAN_1745) simply have some sequence homology with mammalian lipoxygenases. At this point in the manuscript, it has not been defined which enzyme produces 11-HETE.

Answer) Thank you for good checking. As you checked, the identification of the enzymes was not clear. Thus, the content of the identification of the enzymes was newly described as follows: “The protein from *MXAN_5217* converted ARA to PGH₂ (Supplementary Fig. 5), indicating that it is COX. The activity of COX toward ARA was 0.011 μmol min⁻¹ mg⁻¹. In humans, COX converts ARA to PGH₂, which can be converted to diverse PGs by various types of PG synthases (Supplementary Fig. 1). The putative LOX enzymes expressed from *MXAN_1745* and *MXAN_1744*, and the putative EH from *MXAN_1644* were purified from crude cell extracts as single soluble proteins using His-Trap affinity chromatography (Supplementary Fig. 4). The substrate specificity and products of the purified enzymes expressed from *MXAN_1745* and *MXAN_1744*, and *MXAN_1644* are summarized in Supplementary Table 5. The enzymes from *MXAN_1745* and *MXAN_1744* converted ARA to 12-hydroperoxyeicosatetraenoic acid (12-HpETE) and 11-HpETE, respectively, indicating that they are ARA 12-LOX and ARA 11-LOX, respectively. The enzyme expressed from *MXAN_1644* converted HXB₃ to TrXB₃. Thus, it was identified as EH.” **(Line 113–125 in the second revised manuscript)**

9. Line 275, states “...demonstrated PPAR_γ up-regulation”. Only luciferase activity was reported, while expression of PPAR_γ was not evaluated.

Answer) As you checked, we only investigated transcriptional activity of PPAR_γ. Thus, the sentence of “HXB₃, HXB₄, HXD₃, TrXB₃, TrXD₃, and 11-HETE demonstrated PPAR_γ up-regulation.” was revised “HXB₃, HXB₄, HXD₃, TrXB₃, TrXD₃, and 11-HETE increase the transcriptional activity of PPAR_γ.” **(Line 275–276 in the second revised manuscript)**

10. (1) In general, the results with PPAR γ activation are overstated. The authors have simply shown increased activity of the luciferase reporter, as has been demonstrated for numerous other exogenous compounds. This study does not demonstrate direct agonist activity, despite the *in silico* docking studies. (2) Secondary production of other endogenous ligands in these cells cannot be ruled out. (3) Also, in response to my previous comment regarding justification of the concentration used to test for PPAR agonist activity, the authors simply state that “In other reports...high micromolar concentrations were used”. This is not a scientific justification for these particular products.

Answer (1) To describe more exactly the results, the terms of “up-regulation” and “down-regulation” were revised to “increase the transcriptional activity” and “decrease the transcriptional activity”, respectively, and the contents were revised as follows: “Although the increasing degrees of the transcriptional activity of these lipid mediators were less than that of troglitazone (TRO), an antidiabetic and anti-inflammatory drug (Fig. 3a,d,e,h). The PPAR γ activity additively increased when HXB₃ or HXD₃ was supplemented with TRO. The increasing degree of the transcriptional activity of HXB₄ on the transcriptional activity of PPAR γ was similar to that by TRO (Fig. 3b). HXB₅, TrXB₄, and TrXB₅ did not affect the PPAR γ activity (Fig. 3c,f,g). When TRO was supplemented, HXB₅ decreased PPAR γ activity.”

(Line 202–208 in the second revised manuscript)

However, this study does not demonstrate direct agonist activity. To demonstrate exactly agonist activity, we will test whether these lipid mediators show no considerable cytotoxicities, induce adipogenesis, and accumulate lipid droplets. Also, the effects of these compounds on PPAR γ expression as well as PPAR γ -mediated genes will be checked. **(Line 282–285 in the second revised manuscript)**

(2) It has been suggested that the endogenous factor of HEK293 cells do not affect the studies of PPAR γ agonists. Moreover, the lipid mediators tested were not metabolized by HEK 293 cells. Therefore, we think that the secondary production of other endogenous ligands in these cells seemed to be almost ruled out. The related content was described as follows: “Human embryonic kidney (HEK) 293 cells have been widely used for the screening of PPAR γ agonist⁶³, suggesting that the endogenous factor of HEK293 cells do not affect the studies of PPAR γ agonists. The lipid mediators tested were not metabolized by HEK 293 cells (Supplementary Fig. 56).” **(Line 362–365 in the second revised manuscript)** The

metabolites were investigated by HPLC analysis after cultivation of the HEK cells with lipid mediator transfected with plasmids for 24 h. **(Supplementary Fig. 56 in the second revised supplementary information)**

(3) We used the high micromolar concentrations for these assays and justified how these concentrations as follows: “The high micromolar concentration of HETE, HX, or TrXs up to 20 μ M was used to test for PPAR activity assay because 20 μ M was the maximal concentration to show no cytotoxicity. The concentration up to 20–40 μ M has been used for PPAR γ partial agonists due to their weak activities and no cytotoxicities^{59,60,61}. The concentration of the full agonist TRO was the same as those used in other reports^{47,62}.” **(Line 372–376 in the revised manuscript)** The references **(#59, #60, #61, and #62)** were newly cited.

Response to Reviewer #2

1. Lines 268-271: The authors write "Full agonists of PPAR γ such as TMZs are known to bind to H12, whereas partial agonists stabilize the β -sheet and the H2'/H3 area, resulting in distinct transcriptional effects between full and partial agonists. Thus, the synergistic effects of full and partial PPAR γ agonists exist." It is not clear what is meant here. If the mentioned synergistic effect as been observed by others a ref should be added. Unless it has escaped my attention, there is no synergistic effect shown in this paper, at best some additive effects (see results of Fig. 3).

Answer) Thank you for your good comment. We agree the reviewer’s comment that it was not shown exceptional synergistic effect. Because partial agonist can be interacted with full agonist in PPAR γ , the additive effect can be shown. In our study, the partial agonists HXB₃, HXB₄, HXD₃, TrXB₃, TrXD₃, and 11-HETE combined with the full agonist Tro showed additive effect, but not synergistic effect. Thus, we changed the term of “the synergy effect” to “the additive effect”. **(Line 272 in the second revised manuscript)** Moreover, the related reference **(#56)** for the additive effect was newly cited. “Thus, the additive effects of full and partial PPAR γ agonists exist⁵⁶.”

Response to Reviewer #3

1. Title: the title might sound a little bit more concise if it would just read: "Biotransformation of polyunsaturated fatty acids to bioactive hepoxilins and trioxilins by microbial enzymes"

Answer) Thank you for your good comment. We agree the reviewer's comment that changed the title from "Biotransformation of polyunsaturated fatty acids to hepoxilins and trioxilins, human lipid mediators, by microbial enzymes" to "Biotransformation of polyunsaturated fatty acids to bioactive hepoxilins and trioxilins by microbial enzymes". **(Line 1–2 in the second revised manuscript)**

2. (1) The authors rely heavily on the presence of ROESY crosspeaks for determining whether a pair of protons attached to adjacent carbons is in the E or Z (or syn or anti) configuration. I doubt whether this is by itself sufficient, because fig 14a in the suppl. info also shows a crosspeak between H10 and H7, protons that are presumably further separated than H14 and H15 in a hypothetical E configuration. (2) Perhaps the authors can give a reference that demonstrates that ROESY is a suitable method for this purpose. And might coupling constant not also actually help to decipher the actual geometries?

Answer) (1) Thank you for your good check. To describe more exactly the NMR results of HXB₃, the extra text was newly described as follows: "C-7 and C-10 are composed of sp³ bonds, which allow both C-7 and C-10 rotate freely. As H-7 and H-10 are very close to each other in the 3D minimized energy calculated molecular structure of HXB₃, they result in ROE peaks in ROESY NMR." **(Line 160–163 in the second revised supplementary information)**

(2) A double bond is sp², which is structurally fixed. In the case of *E*-geometry of double bond, the protons are 180 degrees apart from each other, so neither NOE peak nor ROE peak does not appear. On the other hand, in the case of *Z*-geometry of double bond, the protons are so close that the ROE peak appears^{1,2}. **(Line 145–149 in the revised supplementary information)** The references **(#1 and #2 in the revised supplementary information)** were newly cited.

3. The authors actually agree with me that only when taking retention times into

account PGs can be identified. Still authors do not carry out such experiment and instead state the metabolite is suggested to be... I do not understand why such a simple experiment is not carried out. It takes half a day to inject authentic standards along with an internal standard and compare relative retention times. At least PGE2 and PGD2 and several other PGs are readily available.

Answer) As suggested, to determine more exactly product of *M. xanthus* COX (MXAN_5217), we carried out by the enzyme reaction with arachidonic acid as a substrate. The reaction product was analyzed through HPLC and compared with the standards PGE₂ and PGH₂. As a result, the reaction product of *M. xanthus* COX (MXAN_5217) showed the same retention time as PGH₂. MS/MS analysis also exhibited that the molecular mass of the reaction product was the same as that of PGH₂. Thus, the reaction product of *M. xanthus* COX was identified as PGH₂. The HPLC result was newly added in supplementary Figure 5b.

The sentence of “COX from *M. xanthus* converted ARA to PG analogues, and its activity toward ARA was 0.011 $\mu\text{mol min}^{-1} \text{mg}^{-1}$ (Supplementary Fig. 5).” was revised to “The protein from MXAN_5217 converted ARA to PGH₂ (Supplementary Fig. 5), indicating that it is COX. The activity of COX toward ARA was 0.011 $\mu\text{mol min}^{-1} \text{mg}^{-1}$.” **(Line 113–115 in the second revised manuscript)** The term of ‘HXs and TrXs’ in the section of HPLC quantitative analysis was revised to ‘HXs, TrXs, and PGs’. **(Line 390 in the second revised manuscript)** The sentence of “The metabolite was suggested as PGH₂.” was revised to “The metabolite was identified as PGH₂.” **(Legend of Supplementary Fig. 5 in the second revised supplementary information)**

4. Manuscript p5 I94. The sentence..."These metabolites did not match against the compounds in the LIPID MAPS database, suggesting that they are newly identified metabolites" does not seem entirely correct to me. In other words, any other sources than LIPID MAPS are left out of consideration, however if a lipid is not in LIPID MAPS doesn't mean i) it has not yet been identified and ii) it has not been published elsewhere.

Answer) Thank you for your careful check. You gave us same comment with reviewer #1. As suggested, we compared new compounds in several databases and research papers besides LIPID MAPS database. Nevertheless, we did not found same compounds with newly metabolites. Thus, the sentence of “These metabolites did not match against the compounds

in the LIPID MAPS database, suggesting that they are newly identified metabolites.” was revised to “These metabolites did not found against the compounds in several information databases, including LIPID MAPS database, PubChem, Human Metabolome Database, and KEGG, and their MS/MS profiles have not been reported elsewhere to date. Thus, they are suggested as new metabolites. ”. **(Line 95–98 in the second revised manuscript)**

Finally, we would like to thank you and the reviewers again for thoughtful suggestions and comments for improvement of this manuscript, and we hope that the second revised version of our manuscript meet the high standards of *Nature Communications* and is now acceptable for publication.

Thank you very much for your interest and assistance.

Sincerely,

Oh

Responses to Reviewers:

Reviewer #1 (Remarks to the Author):

The authors were responsive to my previous comments in their revised manuscript submission. However, there are still some items that should be corrected.

1. Abstract: The line “retaining beneficial anti-diabetic properties with reduced side effects” is completely misleading. As I noted in my previous comments, the authors did not test anti-diabetic effects of these compounds or potential side effects of treatment in vivo. This statement should be removed.

2. Introduction: The statement “However, the produced concentrations of HXs and TrXs are too low...” should be revised. The authors should be more specific (i.e., too low for what?). Also, the next line states that “For the discovery of bioactive compounds as potential drugs, more efficient production and diverse types of HXs and TrXs are required”. As I stated previously, this statement is not clear and has not been improved upon revision. Several lipid mediators have been produced in milligram-gram scale using total organic synthesis approaches and it is not clear why more diverse types are required from a drug development perspective. In my opinion, this type of vague and unsubstantiated language should be justified or removed.

3. Discussion: The new statement “To demonstrate exactly agonist activity, we will test whether these lipid mediators show no considerable cytotoxicity, induce adipogenesis and accumulate in lipid droplets” should be removed. None of these experiments will demonstrate direct agonist activity and the authors should not state which experiments they are planning in the future. They should simply state that direct agonist activity was not determined and leave it at that.

Reviewer #2 (Remarks to the Author):

In this revised version, the authors have satisfactorily answered my query.

Reviewer #3 (Remarks to the Author):

The authors ultimately convinced me that it should be PGH2, the NMR data and its explanation now seem consistent. Nevertheless, this is very hard to judge for me and I can only encourage the authors to be very critical about this. It is a very complex field.

Best regards
Martin Giera

Responses to Reviewers:

Response to Reviewer #1

1. Abstract: The line “retaining beneficial anti-diabetic properties with reduced side effects” is completely misleading. As I noted in my previous comments, the authors did not test anti-diabetic effects of these compounds or potential side effects of treatment in

vivo. This statement should be removed.

Answer) Thank you for your good advice. As you suggested, the sentence of "retaining beneficial anti-diabetic properties with reduced side effects." was removed. **(Line 27 in the third revised manuscript)**

2. Introduction: The statement "However, the produced concentrations of HXs and TrXs are too low..." should be revised. The authors should be more specific (i.e., too low for what?). Also, the next line states that "For the discovery of bioactive compounds as potential drugs, more efficient production and diverse types of HXs and TrXs are required". As I stated previously, this statement is not clear and has not been improved upon revision. Several lipid mediators have been produced in milligram-gram scale using total organic synthesis approaches and it is not clear why more diverse types are required from a drug development perspective. In my opinion, this type of vague and unsubstantiated language should be justified or removed.

Answer) As suggested, the statement of "However, the produced concentrations of HXs and TrXs are too low and their types are limited to less than 10. For the discovery of bioactive compounds as potential drugs, more efficient production and diverse types of HXs and TrXs are required." was removed. The sentence of "Thus, HXs and TrXs are important lipid mediators." was revised to "Thus, HXs and TrXs are important lipid mediators for various organisms." **(Line 53 in the third revised manuscript)**

3. Discussion: The new statement "To demonstrate exactly agonist activity, we will test whether these lipid mediators show no considerable cytotoxicity, induce adipogenesis and accumulate in lipid droplets" should be removed. None of these experiments will demonstrate direct agonist activity and the authors should not state which experiments they are planning in the future. They should simply state that direct agonist activity was not determined and leave it at that.

Answer) As you suggested, the statement of "To demonstrate exactly agonist activity, we will test whether these lipid mediators show no considerable cytotoxicity, induce adipogenesis, and accumulate lipid droplets. Also, the effects of these compounds on PPAR γ expression as well as PPAR γ -mediated genes will be checked." was removed. The sentence of "However, this study does not demonstrate direct agonist activity." was revised to

“However, this study does not demonstrate direct agonist activities of these products.”. (**Line 288 in the third revised manuscript**)

Response to Reviewer #2

1. In this revised version, the authors have satisfactorily answered my query.

Answer) Thank you for your advice so far.

Response to Reviewer #3

1. The authors ultimately convinced me that it should be PGH2, the NMR data and its explanation now seem consistent. Nevertheless, this is very hard to judge for me and I can only encourage the authors to be very critical about this. It is a very complex field.

Answer) Thank you for your good advice for us so far. We agree that the identification of structures is a very complex field. Nevertheless, we clearly identify new materials using LC-MS/MS and NMR in the present. Thus, we think that this identification is meaningful.